# Climate change reduces extent of temperate drylands and intensifies drought in deep soils

Daniel R. Schlaepfer[1,2], John B. Bradford[3], William K. Lauenroth[2,4], Seth M. Munson[3], Britta Tietjen[5,6], Sonia A. Hall[7,8], Scott D. Wilson[9,10], Michael C. Duniway[11], Gensuo Jia[12], David A. Pyke[13], Ariuntsetseg Lkhagva[14] & Khishigbayar Jamiyansharav[15]

Drylands cover 40% of the global terrestrial surface and provide important ecosystem services. While drylands as a whole are expected to increase in extent and aridity in coming decades, temperature and precipitation forecasts vary by latitude and geographic region suggesting different trajectories for tropical, subtropical, and temperate drylands. Uncertainty in the future of tropical and subtropical drylands is well constrained, whereas soil moisture and ecological droughts, which drive vegetation productivity and composition, remain poorly understood in temperate drylands. Here we show that, over the twenty first century, temperate drylands may contract by a third, primarily converting to subtropical drylands, and that deep soil layers could be increasingly dry during the growing season. These changes imply major shifts in vegetation and ecosystem service delivery. Our results illustrate the importance of appropriate drought measures and, as a global study that focuses on temperate drylands, highlight a distinct fate for these highly populated areas.

[1] Department of Environmental Sciences, University of Basel, Section of Conservation Biology, 4056 Basel, Switzerland. [2] Department of Botany, University of Wyoming, Laramie, Wyoming 82071, USA. [3] US Geological Survey, Southwest Biological Science Center, Flagstaff, Arizona 86001, USA. [4] Yale University, School of Forestry and Environmental Studies, New Haven, Connecticut 06511, USA. [5] Freie Universität Berlin, Institute of Biology, Biodiversity and Ecological Modeling, 14195 Berlin, Germany. [6] Berlin-Brandenburg Institute of Advanced Biodiversity Research (BBIB), 14195 Berlin, Germany. [7] Center for Sustaining Agriculture and Natural Resources, Washington State University, Wenatchee, Washington 98801, USA. [8] SAH Ecologia LLC, 669 Crawford Avenue, Wenatchee, Washington 98801, USA. [9] Department of Biology, University of Regina, Regina, Saskatchewan, S4S 0A2, Canada. [10] Climate Impacts Research Centre, Department of Ecology and Environmental Science, Umeå University, 981 07 Abisko, Sweden. [11] US Geological Survey, Southwest Biological Science Center, Moab, Utah 84532, USA. [12] Institute of Atmospheric Physics, Chinese Academy of Sciences, Beijing 100029, China. [13] US Geological Survey, Forest and Rangeland Ecosystem Science Center, Corvallis, Oregon 97331, USA. [14] Department of Biology, School of Arts and Sciences, National University of Mongolia, Ulaanbaatar 210646, Mongolia. [15] Department of Forest and Rangeland Stewardship, Colorado State University, Fort Collins, Colorado 80523, USA. Correspondence and requests for materials should be addressed to D.R.S. (email: daniel.schlaepfer@unibas.ch).

Global climate models (GCMs) project consistent increases of climatological aridity for the twenty first century[1–5]. Yet, GCM projections of meteorological droughts are uncertain and suggest robust increases in some but not all regions[6,7]. This uncertainty could have particularly strong consequences for dryland regions[5,8], which are already limited by water[9,10]. Drylands may respond to climate change in their distribution, driven by aridity, or in ecosystem structure, function, and composition, driven by ecohydrological processes. Global drylands expanded over the twentieth century by 4–8%[2,3] and represent currently c. 40% of the global terrestrial surface (refs 2,5). Despite observations of increasing overall aridity, forecasts of extreme drought events in the second half of the twentieth century remain uncertain[1,4,10,11]. Model projections largely agree, however, that drylands will likely continue to expand during the twenty first century[1–3,5,10] due to increases in evaporative demand and a global hydrological cycle with longer and more severe dry periods[10,12,13]. A net expansion of drylands may reduce ecosystem services and impact human livelihoods[14] through water scarcity[15,16], vegetation die-offs[17] and land degradation[18] all of which are exacerbated by human land use[19]. The projected global trend towards increased aridity is largely robust to variation among models and data sources, even though potential evapotranspiration by itself is unsuitable for understanding drying trends[11,20,21]. However, global temperature and precipitation projections vary geographically and latitudinally[1,10] suggesting different outcomes for tropical and subtropical (hereafter subtropical) drylands versus temperate drylands at mid-latitudes[5]. Of particular concern for dryland ecosystems, trends in meteorological drought and soil moisture are highly uncertain and generally model dependent[6,7].

This uncertainty is especially complicated for soil moisture availability, which is dictated by the combination of weather, vegetation, soil and landscape attributes. In dryland ecosystems, soil moisture controls most ecosystem processes[8,22]. Reduced primary productivity occurs primarily during periods of reduced soil moisture and not directly to an absence of precipitation[8,22,23]. Conditions that diminish harvest yields due to below-normal levels of soil moisture, particularly during the growing period, have traditionally been called agricultural drought (in contrast, for example, to meteorological drought which is a period of below-normal precipitation[8]). The notion of reduced soil moisture has been extended to ecosystems and is referred to as ecological drought[8]. Ecological drought is commonly described as a 'prolonged and widespread deficit in naturally available water supplies [...] that create multiple stresses across ecosystems' (US Geological Survey, US Climate Science Centers and the Science for Nature and People Partnership) and has recently garnered widespread attention as one of the topics defining twenty first century climate change[14]. Because of the complexity of the water cycle, soil moisture and ecological drought projections show large uncertainties among GCMs[1,3,6,7]. Soil moisture projections and drying trends are better constrained in subtropical drylands because these are closely linked to the well-represented Hadley Circulation[1]. Much of the existing research on climate change impacts to drylands has focused on climatic aridity and meteorological droughts, or has been restricted to subtropical drylands. As a result, much less is known about impacts of climate change on soil moisture and ecological droughts, and in particular in temperate drylands.

Vegetation responds to and influences soil moisture through transpiration, interception, shading, and hydraulic redistribution[8]. Despite adaptations of dryland vegetation to ambient aridity levels[8,24], responses to increased droughts and warming under climate change remain difficult to constrain. Potential outcomes include plant functional type shifts[18,25],

woody plant mortality[17] and encroachment[26], and resistance of some vegetation types[24]. These vegetation responses vary among plant functional types and depend on seasonal and soil depth dynamics of soil moisture in addition to climate[8,22,27]. Three plant functional types—shrubs, $C_3$ grasses and $C_4$ grasses—most frequently dominate temperate dryland vegetation. While all types use shallow soil moisture, shrubs can water from greater depths[8,22]. Shifts in the relative dominance of plant functional types, particularly those involving woody species, can impact ecosystem water balance by altering water uptake and evapotranspiration[26]. Woody plant encroachment has been a concern in grass-dominated drylands worldwide during the twentieth century and is projected to increase under climate change[26]. Changes in vegetation in response to changes in soil moisture may impact ecosystem services in temperate dryland ecosystems globally.

We applied a two-tiered approach to assess consequences of climate change for global temperate, arid and semiarid drylands. First, we quantified zones of contraction, expansion and stability of the distribution of five temperate dryland regions. Second, we estimated impacts of climate change on seasonal and depth patterns of ecological drought, and their consequences for plant water uptake using SOILWAT[28,29], an ecosystem water balance simulation model. SOILWAT utilizes site-specific soils and weather data (here we evaluated spatially and temporally downscaled output from 16 GCMs driven by an intermediate and a high emissions scenario), and SOILWAT soil moisture outputs compare very favourably with GCM estimates (see Methods). Furthermore, SOILWAT provides high temporal resolution (daily) information about ecosystem water balance and plant available moisture that reflects the influence of site-specific soil conditions.

Here we illustrate that GCMs for the late twenty first century project a net loss of c. 15% (following the representative concentration pathway (RCP) 4.5 (ref. 1)) to 30% (following RCP8.5) of current temperate dryland extent due to climatic changes. We show that the duration of ecological droughts during growing periods may substantially increase, especially in deeper ($>20$ cm) soils. Water uptake by vegetation under future climate could be increasingly reliant on surface soil moisture, favouring shallow-rooted over deep-rooted vegetation, which contrasts with previous projections of increasing dryland woody encroachment[26]. Plant water uptake patterns within and among regions are projected to become more similar, suggesting a homogenization of niche spaces and vegetation composition. Our findings emphasize contrasting spatial trajectories between subtropical and temperate drylands and highlight the need to assess seasonal as well as spatial patterns of soil moisture dynamics to understand factors that shape the future of temperate drylands and the services they provide.

## Results

**Spatial response of temperate drylands to climate change.** The extent of temperate drylands under current climate is $8.3 \times 10^6$ km$^2$ based on aridity, climate zone, and mean annual temperature (MAT) (Fig. 1 and Supplementary Table 1). This corresponds to c. 5.6% of the global terrestrial surface and to 20–30%, varying by published estimates[2,5], of all arid and semiarid areas globally. Changes in aridity, climate zone, and mean annual temperature projected by GCMs will alter the future distribution of temperate drylands, which we defined here climatologically[2]. By the end of this century, climate change could lead to a net contraction of temperate drylands of up to $2.4 \times 10^6$ km$^2$ ($1.2$–$3.3 \times 10^6$ km$^2$ among 16 GCMs following RCP8.5) with considerable variation among regions (Fig. 1 and

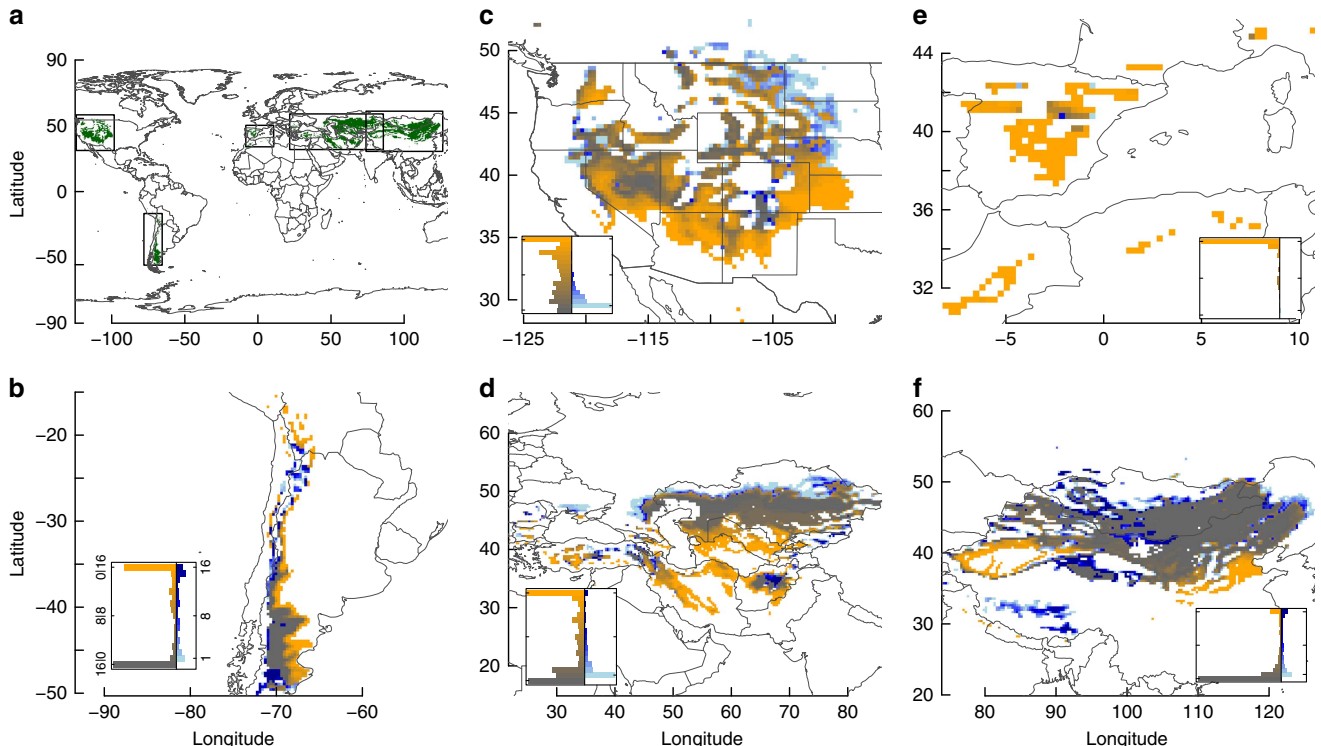

**Figure 1 | Current and future distribution of temperate drylands.** (**a**) Five temperate dryland regions with their current extent for 1980–2010 (green): (**b**) South America; (**c**) North America; (**d**) Western and Central Asia; (**e**) Mediterranean Basin; (**f**) Eastern Asia. (**b**–**f**) Future projected change in extent under RCP8.5 for 2070–2100, depicted as stable (grey), contracting (orange; no longer temperate dryland in 2070–2100), and expanding (blue; newly temperate dryland in 2070–2100) zones. Inset vertical histograms for **b**–**f** illustrate areal abundance in each category of GCM agreement about expansion or contraction of temperate drylands. Left (grey-orange) histogram depicts GCM agreement (that is, number of GCMs that agree in the direction of change) about the fate of current temperate drylands and shows the number of cells within each category ranging from pure grey (all 16 GCMs forecast stable temperate drylands) to pure orange (all GCMs forecast conversion from temperate dryland to non-temperate and/or non-dryland). Right (light blue—dark blue) histogram indicates GCM agreement of temperate dryland expansion into new areas and shows the number of cells within each category ranging from dark blue (all GCMs forecast conversion to temperate dryland) to light blue (one GCM forecasts conversion).

Supplementary Fig. 1 and Supplementary Table 2). RCP8.5 represents a 'business as usual' scenario, that is, no mitigation to curb climate change, which will not occur if the Paris agreement[30] to keep the global mean temperature 'well below 2 °C above pre-industrial levels' is implemented. All results for the intermediate emissions scenario RCP4.5, which assumes a stabilization of emissions without overshoot, are given in Supplementary Figs 1–10 and Supplementary Tables 1–3, 5, 7, and 10, but are qualitatively similar. While other studies indicate that drylands in total may increase by 5–23% globally[2,5], that general statement masks our result that temperate drylands may contract while subtropical drylands expand. We found that a median of 36% (24–51% among GCMs) of current temperate drylands would be converted under the considered scenario mainly to warmer subtropical drylands (Supplementary Table 3). An area equal to 9% (6–20%) of the current extent would be added in the future as temperate drylands, primarily because of increased aridity in currently sub-humid areas (Supplementary Table 3). Our assessment of contracting, stable, and expanding zones among GCMs showed consistency in four regions (32–80% agreement), but not in North America (19%; Fig. 1b–f insets).

**Duration and distribution of ecological droughts.** Ecological droughts during growing periods, which we estimated as the longest snow-free, frost-free period when soil water potential was continuously < − 3.0 MPa, could last longer under projected future scenarios (Fig. 2). Our model, driven by soil data and

climate inputs from 16 GCMs, projected increasing drought periods in every temperate dryland region, except for parts of Asia, that are not projected to shift in distribution under climate change (Fig. 2 and Supplementary Tables 4–5). Ecological droughts may become longer over 65% (31–96% among GCMs) of the area of temperate drylands in surface soil layers (0–20 cm) and 85% (68–97%) in deeper layers (> 20 cm). This increase in growing season droughts coincided with a reduction of the warm/wet season overlap due to increasing cold-season precipitation (Supplementary Figs 2–6 and Supplementary Tables 6–7). Increasing ecological drought, particularly during the warm and dry season[13], is consistent with other evaluations[1–4], and will have consequences for dryland vegetation, including elevated plant mortality, more frequent wildfires, and shifts in plant functional types[8,17,19,22,23]. East Asia is the only region with projections that consistently diverged from the trend of increasing ecological drought, which is consistent with previous studies[1]. This may be related to East Asia being the only region with a positive warm/wet season overlap (Supplementary Fig. 5). Ecological droughts in East Asia may become shorter instead of longer in over 43% (surface layers) and 26% (deeper layers) of the region.

The projected intensification of ecological droughts is more pronounced for deep layers (+10%, 0–20%, corresponding to +18 days, 8–38 days, longer dry periods) than surface layers (0%, −12 to 30%; +2.6 days, −7 to 17 days) particularly for contracting and expanding zones. This result was surprising since increased cold-season precipitation might be expected to enhance

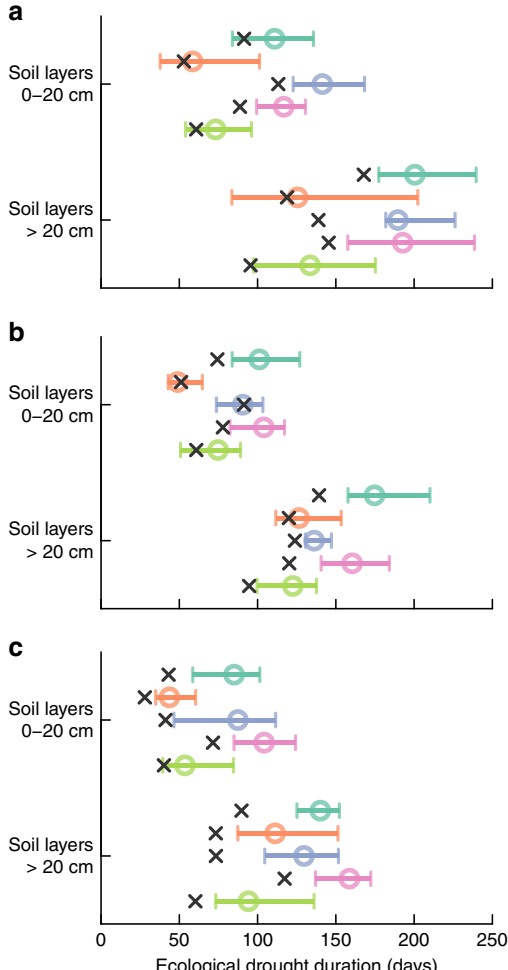

**Figure 2 | Duration of ecological droughts during growing season.** Soil drought (days) shown under current (black cross) and future projected conditions under RCP8.5 for 2070–2100 in contracting (**a**), stable (**b**) and expanding (**c**) zones of temperate dryland regions. Error bars (horizontal lines) represent minimum–maximum range of values among 16 GCMs and median (circle) represent each region (Fig. 1; turquoise, South America; orange, Eastern Asia; purple, Western & Central Asia; pink, Western Mediterranean; green, North America). We estimated the duration of ecological droughts during growing periods as the longest snow-free, frost-free period when soil water potential (SWP) $< -3.0$ MPa continuously.

available soil moisture at depth due to reduced evaporative competition for percolating soil moisture[8,22]. As a consequence of differential drying of deep versus surface soil layers, future vegetation was projected to extract more water from shallow rather than deeper layers. Our simulations suggest that overall the importance of transpiration from shallow layers increases under climate change, that is, the proportion of total transpiration that derives from deep layers decreases (Fig. 3 and Supplementary Tables 4–5). We estimated the proportion of transpired water derived from deep soil layers to decrease by a median of 8% (4–12%) for South America, 2% ($-2$ to 7%) for Central and Western Asia, 11% (7–15%) for the Western Mediterranean basin, and 5% ($-1$ to 9%) for North America. The exception was again East Asia where we estimated increased water uptake (2%, $-2$ to 5%) from deep soil layers (Supplementary Fig. 7 and Supplementary Tables 4–5). Our simulation results suggest also that transpiration from shallow layers may increase in the median case in most regions. Median decreases occur in the Western

Mediterranean basin and the expanding zone of South America. In addition, our results also indicate a heterogeneous pattern where the overall regional trends are interrupted at smaller spatial scales (Supplementary Fig. 8 and Supplementary Tables 4–5). This heterogeneous pattern of the geographic distribution of increases and decreases is more prominent for transpiration derived from soil moisture at deep soil layers (Supplementary Fig. 9 and Supplementary Tables 4–5). Among regions and within some regions (specifically East Asia, South America and the Western Mediterranean), we found a negative relationship between the projected change in the proportion of transpiration derived from deep soil moisture and the current value (Fig. 4). This negative relationship indicates a homogenization of plant water uptake among soil layers implying a reduction of niche spaces, associated plant functional types, and biodiversity[8,22] within temperate drylands as a whole and within those regions that display the negative relationship (Fig. 4).

**Discussion**

Net reductions in the area of temperate drylands occurred in our projections following an intermediate and a high-emission scenario across all five temperate dryland regions and illustrate the different impact of climate change on the distribution of temperate versus subtropical drylands. The latter are projected to expand due to conversions from temperate to subtropical climate in addition to increased aridity in currently sub-humid subtropical regions[1,3,5,10]. Consequences for vegetation of a shift from temperate to subtropical drylands include loss of temperature-controlled seasonal cycle, phenological shifts, increases in frost-intolerant species and dominance of $C_4$ over $C_3$ grasses. Furthermore, impacts on ecosystem services could have large consequences for human well-being: aggressive human diseases, including dengue and schistosomiasis[31], as well as mound-building termites[32], occur in subtropical climates and could expand as temperate drylands warm, whereas cool season crops such as wheat and potato would no longer be economically viable[33].

Our findings suggest large and regionally variable shifts in the distribution of temperate drylands under a changing climate, and highlight the complex interplay between seasonal soil water resources and intensified ecological droughts during the growing season that differ with soil depth. While increased water availability at depth would be expected with more cold-season precipitation (favouring woody and deep-rooted species[8,22]), our results suggest instead a soil moisture regime that is increasingly dominated by longer ecological droughts particularly at depth and by available water restricted to surface soils (favouring shallow-rooted herbaceous species[8,22]) and the cool season (favouring winter annuals, including invasive grasses[34]). Increasing water scarcity in deep soils is relevant for ecosystem function because soil moisture at depth is an important resource for deep-rooted woody species as drought proceeds[27]. This indicates, for instance, that expected future increases of woody plant encroachment[26] may not be generalizable across all drylands. Our study emphasizes the need to differentiate among drylands and describes intensifications of seasonal and soil depth patterns of drought that could affect temperate dryland plant communities and the services they provide, including water resources, wildlife habitat, soil conservation, agriculture and carbon storage.

**Methods**

**Study area.** We identified temperate drylands using three criteria: mean annual temperature (MAT), the Trewartha climate classification scheme[35], and the FAO/UNEP aridity index (AI) (ref. 36). In addition, we restricted the analysis to areas with soils of less than 90% sand content. We classified temperate areas if

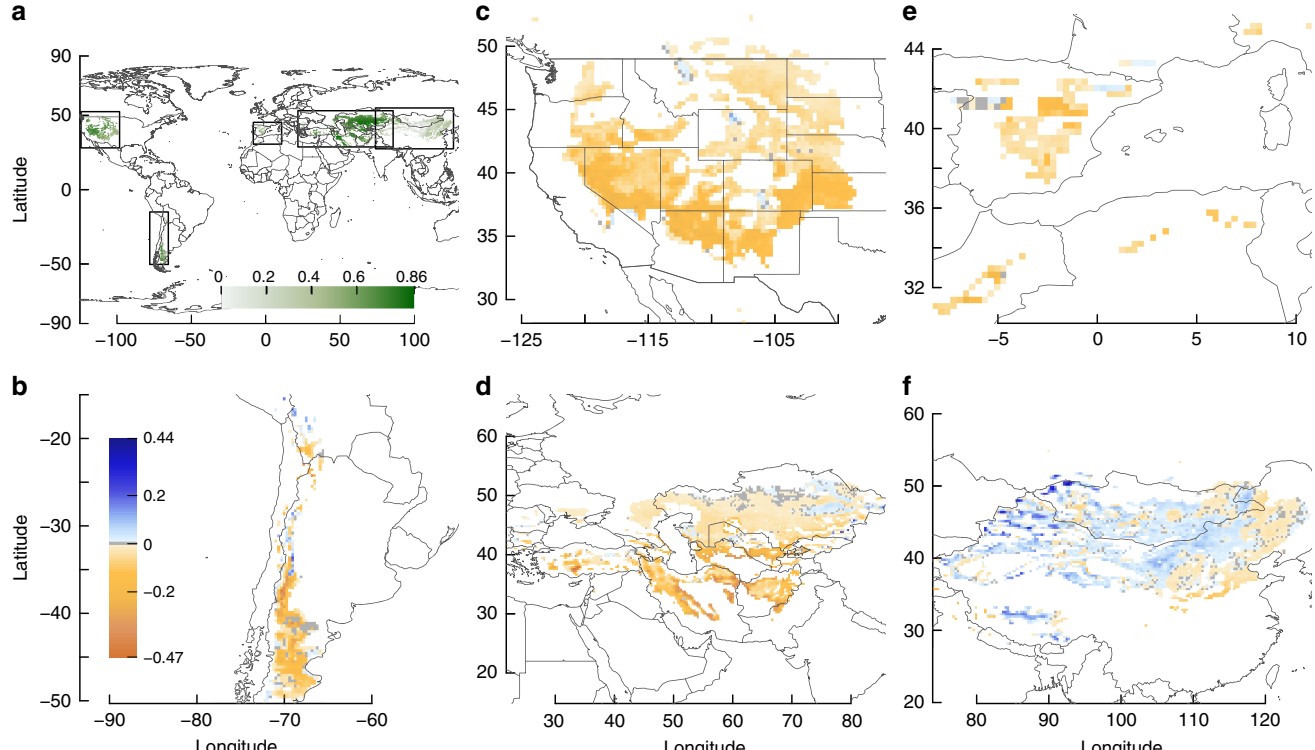

**Figure 3 | Climate-driven changes in the proportion of transpiration derived from deep soil moisture.** (**a**) Current values; dark green indicates areas with transpiration primarily from deep layers, >20 cm depth. (**b–f**) Impact of climate change in the study regions, expressed as the difference in the proportion of transpiration derived from deep layers between future consensus projections under RCP8.5 for 2070–2100 and current conditions. Dark orange indicates decreasing proportion of transpiration from deep layers, dark blue indicates increasing, and grey indicates no change. Areas depicted include all cells that are either current and/or future temperate drylands under any GCM (Fig. 1).

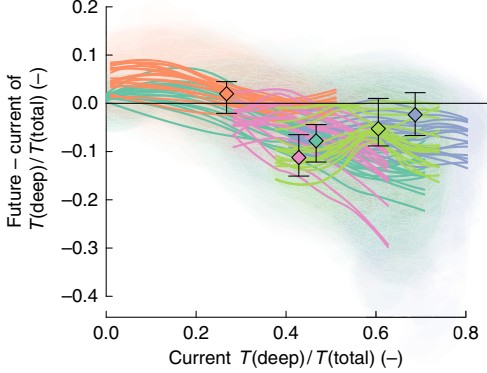

**Figure 4 | Relationship of the proportion of transpiration derived from deep soil moisture between current values and its future response.** Current values refer to transpiration derived from soil moisture at >20 cm depth; future response refers to the change between values under RCP8.5 for 2070–2100 and current values. Regional responses are summarized for each GCM by locally weighted polynomial regressions (lines) and 90% data clouds (shaded areas) for all areas that meet either current and/or future classification (Fig. 1; turquoise, South America; orange, Eastern Asia; purple, Western & Central Asia; pink, Western Mediterranean; green, North America). Coloured diamonds are median GCM for each region and error bars indicate the GCM range.

MAT >0 °C and Trewartha climate group was category D, that is, the number of months with mean temperatures ≥10 °C is ≥4 but <8. We included areas as drylands if 0.05≤AI<0.5, that is, arid and semiarid types excluding hyper-arid[37]. AI was calculated as MAP/PET, where PET is potential evapotranspiration and MAP is the mean annual precipitation[37], particularly, this index is equivalent to the inverse aridity index of the Budyko framework[20]. Because our classification is

climate dependent, we determined the study area under current climate and for each future climate scenario.

We applied a geographic raster with 0.3125° square cells, so that exactly one cell centre of the NCEP/CFSR T382 Gaussian grid[38] fell in each of our cells. Our raster contained 1,152 × 576 cells and had its origin at 90° S and 179.84375° W. We made an initial generous estimate due to lack of complete knowledge about which cells may be identified as temperate drylands. From the total possible 663,552 cells in the raster, we included 20,021 cells for running simulations. After completing simulation runs, we determined that 12,638 out of the 20,021 raster cells classified as temperate drylands under either current climate or at least one future scenario. We considered only this subset of cells for further analysis.

We grouped the 12,638 raster cells in six geographic regions (Fig. 1) based on the UN geoscheme (United Nations Statistics Division: Composition of macro geographical (continental) regions, geographical sub-regions, and selected economic and other groupings; available at http://unstats.un.org/unsd/methods/m49/m49regin.htm; accessed on 4 Feb 2014). 'South America' (<15° N & >25° W); 'Southern Africa' (<0° N & (>0 & <55)° E)—we omitted Southern Africa from further analysis because only one cell under a few climate conditions was identified as temperate dryland; 'Eastern Asia' including the eastern portion of Southern Asia (along border of Afghanistan/Pakistan except area around city of Quetta) and the eastern portion of Eastern Europe (>87° E starting about at the border point of Russia, Kazakhstan, and China); 'Western and Central Asia' including the western portion of Southern Asia (along border of Afghanistan/Pakistan plus area around city of Quetta) and western portion of Eastern Europe (<87° E); 'Western Mediterranean basin' (W of the Dinaric and Pindus Mountains) including Europe and Northern Africa, but excluding Eastern Europe (>0° N and (<25° W and <14° E); 'North America' (>25° N and >50° W).

**Simulation framework.** We utilized SOILWAT, a daily time step, multiple soil layer, process-based, simulation model of ecosystem water balance[28,29,39]. SOILWAT has been applied and validated in dryland ecosystems including temperate grasslands[28,40], temperate shrub-dominated ecosystems[29] and temperate dry-domain forests[41]. Inputs to SOILWAT include daily weather conditions (mean daily maximum and minimum temperature and daily precipitation), mean monthly climate conditions (mean monthly relative humidity, wind speed and cloud cover), latitude, elevation, vegetation (mean monthly live, standing and litter biomass, active root depth profile) and soil properties (texture of each soil layer). SOILWAT estimates processes for each functional plant group including

interception by vegetation and litter, evaporation of intercepted water, transpiration and hydraulic redistribution from each soil layer. Transpiration and evaporation are estimated by limiting potential rates with stress functions of soil water potential, atmospheric demand, seasonal leaf area, rooting distribution, vegetation-specific critical soil moisture values of water extraction and shading of canopy and litter[41]. This is an approach comparable to the modified Jarvis–Stewart model[42,43]. SOILWAT estimates hydrological processes including partitioning of precipitation into snowfall and rain, snow accumulation, melt and loss, infiltration into the soil profile, percolation for each soil layer, bare soil evaporation and deep drainage[29,39]. PET is calculated using the Sellers' formulation[44] of Penman[45] which incorporates day length effects. Because estimations of PET with Penman-based equations using data from NCEP/CFSR tend to under-estimate PET in dry regions[46], we corrected our PET estimates by multiplication with 1.2 based on a comparison with PET values for 1961–1990 (FAO global map of monthly reference evapotranspiration—10′; available at http://www.fao.org/; accessed on 24 Oct 2012)

Our simulation experiment consisted of a total of 20,020 cells, which we subjected to present climate and two RCPs (RCP 4.5 and RCP 8.5) and the resulting climate projections of 16 global circulation models. We executed this experiment on Yellowstone at the National Center for Atmospheric Research-Wyoming Supercomputing Center[47] and Advanced Research Computing Center's Mount Moran/Bighorn facilities at the University of Wyoming.

**Input data for weather conditions and climate scenarios.** We used NCEP/CFSR products[38] on a T382 Gaussian grid (resolution of ~0.312° × ~0.312°) to simulate current climate conditions (1979–2010; Supplementary Figs 2–6 and Supplementary Tables 6–7). Specifically, we extracted daily maximum and minimum temperature (2 m above ground) and precipitation from the 6-hourly data sets (ds093.0 and ds093.1 (ref. 48)). We also extracted relative humidity (2 m above ground), u- and v-wind speed components (10 m above ground), and total cloud cover, which we converted to sky cover via sunshine per cent[49] from the monthly data set (ds093.2 (ref. 48)) and calculated mean monthly values.

We extracted for the centre of each cell 32 projected future climate conditions as monthly time-series for 2069–2099 from 1/2° downscaled and bias-corrected products of the fifth phase of the Climate Model Intercomparison Project[50] (CMIP5) of 16 global circulation models (GCMs) for two RCPs[51], RCP4.5 and RCP8.5, from the 'Downscaled CMIP3 and CMIP5 Climate and Hydrology Projections' archive[52] at http://gdo-dcp.ucllnl.org/downscaled_cmip_projections/ (data accessed on 4 Feb 2014). We combined historical daily data (NCEP/CFSR) with monthly GCM projections of historical and future conditions with a hybrid-delta downscaling approach to obtain future daily forcing[53,54]. We selected 16 GCMs from all those that participated in CMIP5 that represented the most independent and best performing subset of GCMs[55] (Supplementary Table 8).

Changes in annual precipitation across temperate drylands showed an overall median increase of $+48$ mm yr$^{-1}$ ($-13$ to 91 mm yr$^{-1}$); however, there was important variation among regions ($-39$ mm yr$^{-1}$ for South America to $+58$ mm yr$^{-1}$ for Western and Central Asia) as well as within regions (Supplementary Fig. 2 and Supplementary Tables 6–7). MAT increased consistently across GCMs by $+5.2\,°C$ ($3.4–7.3\,°C$) for all regions except South America, which experienced the lowest increases of $+3.1\,°C$ (Supplementary Fig. 3 and Supplementary Tables 6–7). PET increased similarly consistent with an overall median of $+151$ mm yr$^{-1}$ ($94–209$ mm yr$^{-1}$; Supplementary Fig. 4 and Supplementary Tables 6–7). The typical precipitation regime under current conditions was dominated by cold-season precipitation except for Eastern Asia, which showed consistent warm-season precipitation. Winter precipitation is the sum of mean monthly precipitation of December, January and February on the northern hemisphere, and the sum of mean monthly precipitation of June, July and August on the southern hemisphere. Wet/warm-season overlap is the mean annual Pearson correlation coefficient between mean monthly temperature (°C) and monthly precipitation (mm). $+1$ indicates a perfect match between the warm and wet season; $-1$ indicates a perfect match between the cold and the wet season. Median changes in the overlap of the wet/warm-season were for each region (except South America) and overall mostly small with a trend towards less wet/warm-overlap by $-0.019$ ($-0.064$ to $0.045$), but varied within regions and among GCMs from more wet/cold- to more wet/warm-season overlap (Supplementary Figs 5–6 and Supplementary Tables 6–7).

**Input data for soil characterization.** Soil texture data were derived from the ISRIC-WISE global soil data set[56] at 5 arc-min spatial resolution and at 20 cm depth intervals up to 1 m. We split the 0–20 cm layer into two 10 cm deep layers to improve the representation of surface soil processes and account for ISRIC-WISE cells of 10 cm depth (lithosols). We calculated sand and clay content for each layer and cell as area-weighted averages of soil map units and soil types. Soil depth was based on the ISRIC-WISE data set unless the soil was deeper that 1 m, in which case depth was estimated as 95% of the maximum root depth with 50 cm depth intervals[57] and soil texture was assumed to be the same as the deepest ISRIC-WISE layer. We calculated elevation for each raster cell as area-weighted median based on a 30-arcsec global elevation data set[58].

**Model representation of vegetation.** We assumed that a potential vegetation characterized by three functional groups, shrubs, $C_3$ grasses and $C_4$ grasses,

sufficiently described ecohydrological processes including transpiration, water extraction by roots, timing of water use (phenology) and hydraulic redistribution. Potential vegetation composition of the three functional groups, mean monthly biomass, litter and phenology were based on climate relationships and calculated for each cell and climate condition (detailed description in Bradford et al.[59]). Vegetation composition estimates were based on Paruelo and Lauenroth[60] with an adjustment for the $C_4$ grass component based on Teeri et al.[61] Cell- and climate-condition-specific precipitation modulated mean monthly above ground total, live and litter biomass and temperature-modulated growing season timing and length for each functional group. Rooting depth distribution for each functional group was based on a reanalysis of a global root data set[62] using equations by Jackson et al.[63] for our study area.

Our simulation model made the simplifying assumption that net, ecosystem-scale water-use efficiency and net primary productivity do not respond to increasing atmospheric $CO_2$-concentration. This assumption may under-estimate the effects on soil drying[11,15,21]; yet several studies find a negative net effect of elevated $CO_2$ on soil moisture in dry ecosystems[64,65] and in combination with nutrient limitation[66,67]. Increases in leaf-level water-use efficiency may lead to a positive response in biomass and transpiration in water-limited systems (as opposed to energy-limited systems) and thus, potentially, to a decrease in soil moisture over the long run[64,65,68]. Several major issues remain to be addressed for an accurate representation of responses and their interactions to increasing $CO_2$-concentration at ecosystem scales in models[69–71] before these simulation models will be able to represent the full range of experimental and historical observations[68,72–75]. The importance of these uncertainties is illustrated by the large range of reported values from observations and experiments. These indicate that responses to elevated $CO_2$-concentration of ecosystem water-use efficiency may range from 0 to $+120\%$, of transpiration from $-14$ to $+11\%$, of productivity from 0 to 40%, and of soil moisture from $-20$ to $+10\%$. While the physiological response to $CO_2$-concentration of photosynthesis and leave-level water-use efficiency are reasonably understood[67,76], our ability to predict net impacts at the ecosystem-scale has been described as 'very low'[76].

**Analysis of response variables.** Each SOILWAT simulation run produced daily output for each process and water compartment for the 31-year simulation period discarding the first year as spin-up (see 'Simulation framework'). On the basis of the daily data, we calculated derived response variables (see next paragraph) and then aggregated temporally across 31 years by mean and standard deviation. We calculated these derived and aggregated variables for the current climate condition and for 16 GCMs under two RCPs. We captured the variation among GCMs for each RCP by agreement level of temperate dryland classification and by the selection of study area cells for the aggregation of response variable values (details in 'Variation of response variables'). Because our simulation experiment was deterministic, we estimated effect sizes and performed an evaluation of simulation results, but no statistical hypothesis testing[77]. We used R version 3.1.2 (ref. 78) for all analyses and for creating figures; we used the R package 'maps' version 3.0.2 to add country borders to figures of geographic data.

We chose two derived response variables to capture ecohydrological constraints on potential vegetation. We estimated the mean annual duration of continuous ecological drought during growing periods for surface soil layers of 0–20 cm depth (DDGP0) and for deep soil layers $> 20$ cm depth (DDGP20) as the longest snow-free, frost-free period when soil water potential (SWP) $< -3.0$ MPa continuously. We estimated mean annual proportion of transpiration derived from deep soil moisture ($> 20$ cm depth; T20/T) as the ratio of transpiration resulting from water uptake from deep soil layers (T20) to transpiration resulting from water uptake from all soil layers (T).

**Variation of response variables.** We allowed for variation among raster cells within regions, variation among regions, variation among RCPs, and variation among GCMs. Here, we reported results under RCP8.5, which is closely tracked by recent greenhouse gas emissions[79]. However, RCP8.5 represents a 'business as usual' scenario without mitigation; if the Paris agreement[30] to keep the global mean temperature 'well below 2 °C above pre-industrial levels' is implemented, results under RCP4.5 (Supplementary Information) or RCP2.6 (not simulated) could be more realistic. In the article, we focus on variation among regions and among GCMs (note: overall variation among RCP was for precipitation-related variables as large as variation among GCMs, Supplementary Table 9). The variation among GCMs arose due to spatial variation of extent and location of our study area (temperate drylands are defined as a function of climate) and due to within-cell variation in forcing from the 16 GCMs.

We estimated level of spatial agreement by counting GCMs that classified a cell as temperate dryland. We identified three shifting zones for each GCM: the contracting zone comprises cells with a temperate dryland under current, but not under future climate condition; the stable zone comprises cells with a temperate dryland under current and future conditions; the expanding zone comprises cells with a temperate dryland under future, but not current conditions. We calculated summaries by region and shifting zone in two steps to simultaneously account for both aspects of variation among GCMs, that is, the within-cell and the spatial components. We first calculated for each GCM the target summary statistic among those cells that are part of a zone and region. In a second step, we calculated the

median value among the 16 GCM summary values and used the minimum–maximum GCM range as indicator of among GCM variation. We determined for each shift the contribution of each defining factor and determined whether a cell changed the climate classification between temperate and boreal[35] (1–3 months with mean temperature $\geq 10\,^\circ$C), subtropical[35] ($\geq 8$ months with mean temperature $\geq 10\,^\circ$C) or tropical[35] (12 months with mean temperature $\geq 18\,^\circ$C), and whether a cell changed the aridity classification[36].

We estimated the relative contributions of cells, regions, shifting zones, GCMs, and RCPs to the variation of two groups of variables: climate inputs/drivers (MAT, MAP, AI, PET, wet/warm-season overlap) and the derived ecohydrological response variables. We calculated the uniquely attributable variation based on additive elements by Whittaker[80,81] as percentage of the total variation for each variable for the extent of the study area for each climate condition. We partitioned the variation for the absolute variable values under each climate condition and as difference between future and current conditions. Absolute values indicated that most of the variation was attributable to among-cell variation (mean $\pm 1$ s.d. are $68 \pm 30\%$ for climate drivers, and $63 \pm 19\%$ for response variables) and most of the rest to among-region variation (Supplementary Table 9). Variation attributable to among-GCM and among-RCP variation were of similar size, but only relevant for differences between future and current conditions ($20 \pm 11\%$ and $22 \pm 27\%$ for climate drivers, and $9 \pm 5\%$ and $7 \pm 12\%$ for response variables). The large variation among climate drivers for the attribution of RCP arose because MAT and PET were primarily driven by variation in RCP whereas other drivers (MAP, AI, wet/warm-season overlap) showed larger variation among GCMs.

**Comparison of SOILWAT results with other approaches.** We compared projections of GCMs against SOILWAT output of mean monthly soil moisture. The variable 'mrso' (total soil moisture content) was extracted for seven GCMs under historical and future (RCP4.5 and RCP8.5) scenarios from non-downscaled data from the ESGF node https://pcmdi.llnl.gov/. We calculated normalized mean monthly values for the periods of 1980–2005 and 2070–2099 for each of our simulated raster cells and compared agreement with equivalent soil moisture values from SOILWAT output. We estimated agreement between models with Duveiller's $\lambda$, which is the best performing symmetric agreement index[82]. $\lambda$ ranges between 0 and 1 where 0 indicates no agreement and 1 is perfect agreement. $\lambda$ is proportional to Pearson's correlation index and accounts for systematic and unsystematic bias.

The comparison is favourable with an overall agreement level for the historical time period of $0.89 \pm 0.07$ (mean $\pm$ s.d. among 7 GCM-SOILWAT comparisons) as well as for the future time period under RCP8.5 of $0.92 \pm 0.04$ (Supplementary Fig. 10 and Supplementary Table 10). Regional agreement is mostly similarly high. GCM-SOILWAT agreement, however, was low for Eastern Asia in the historic time period with $0.37 \pm 0.21$, which increased to $0.67 \pm 0.09$ for the future period under the RCP8.5 scenario. Our simulations for the historic time period were run with observation-based weather data, whereas the GCM output represents hindcasts. For the future time periods, the representation of climate conditions for our simulations were based on GCM output. Thus, we expected a higher agreement between our simulation results and those from GCMs for the future time periods than for the historic period. Freedman et al.[83] compared GRACE satellite observations of terrestrial water storage with GCM predictions for 2003–2012 for the Mississippi River Basin and found good agreement in overall aggregated values, but considerable GCM deviations spatially and in water flux partitioning. In a similar exercise, Wu et al.[13] compared GRACE data to GCM predictions to select GCMs for a hydrological impact assessment and found noticeable variation among GCM soil moisture predictions including GCMs with cycles that do not match the seasonal variation. It is not surprising that we found modest deviations between SOILWAT and GCM soil moisture values as well (Supplementary Fig. 10 and Supplementary Table 10), particularly across Eastern Asia, which is a region where several GCMs demonstrate difficulties in representing the monsoonal precipitation regime[84].

We compared SOILWAT output to the demand-supply relationship of water availability in the Budyko framework. We fitted annual output of SOILWAT for $F = E_{actual}\,/\,P$, that is, the ratio of actual evapotranspiration (AET $= E_{actual}$, mm) to annual precipitation (MAP $= P$, mm), against the Budyko aridity index $AI_b = 1/AI_{UNEP}$, that is, the ratio of potential evapotranspiration (PET $= E_{potential}$, mm) to annual precipitation[36]. We used Fuh's equation to represent the Budyko curve, that is, $F = 1 + AI_b - (1 + AI_b^\omega)^{1/\omega}$ (ref. 85); while $AI_b$ describes the prevailing climatic conditions, $\omega$ can be interpreted as the combined influence on water availability of all other factors such as vegetation, soil and seasonality[86]. We estimated $\omega$ for each region based on mean annual SOILWAT output of AET and PET by numerically minimizing the sum of the squared differences between $F$ and $1 + AI_b - (1 + AI_b^\omega)^{1/\omega}$ (refs 86,87) across simulated cells.

The resulting Budyko curves agree well with SOILWAT output that was aggregated without functional constraints, for example, when described by locally weighted polynomial regression lines (Supplementary Fig. 10). We find this favourable agreement with the Budyko framework despite the fact that our estimates of $\omega$ are not precise because we used mean annual values (that is, spatial instead of temporal variation) and because our simulations included only temperate drylands, for example, cells with $AI_b < 2$ are mostly not included (but would contain most of the information of the shape of the curves). This comparison confirms that that actual evapotranspiration in dry regions is limited

not primarily by potential evapotranspiration rates, but by other factors including seasonal soil moisture, soil conditions, and vegetation[20,39]. Understanding climate change impacts in dryland regions, thus, requires models such as SOILWAT, which simulate these 'other' factors in detail and which do not rely on an aridity index as model driver[11,20,21].

**Code availability.** The source code of SOILWAT v0.1.0-gtd is available from our github repository as R package https://github.com/Burke-Lauenroth-Lab/Rsoilwat and the code to run this simulation experiment as R script v0.1.0-gtd from https://github.com/Burke-Lauenroth-Lab/SoilWat_R_Wrapper. The R scripts used to analyse the simulation output are available from https://github.com/drschlaep/GTD_vulnerability.

**Data availability.** The data (simulation outputs) that support the findings of this study are available from the John Wesley Powell Center for Analysis and Synthesis (https://doi.org/10.5066/F7930RB1).

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

## Acknowledgements

This work was conducted as a part of the Climate Change and Ecohydrology in Temperate Dryland Ecosystems: A Global Assessment working group supported by the John Wesley Powell Center for Analysis and Synthesis, funded by the US Geological Survey. We thank the University of Wyoming for additional funding and computational resources. We thank Ryan Murphy for help with programming and data management. We acknowledge the World Climate Research Programme's Working Group on Coupled Modelling, which is responsible for CMIP, and we thank the climate modeling groups (listed in Supplementary Table 8) for producing and making available their model output. For CMIP the U.S. Department of Energy's Program for Climate Model Diagnosis and Intercomparison provides coordinating support and led development of software infrastructure in partnership with the Global Organization for Earth System Science Portals. The use of any trade, product or firm name is for descriptive purposes only and does not imply endorsement by the US Government.

## Author contributions

D.R.S., J.B.B. and W.K.L. designed the study with the help of all authors. J.B.B. with the help from W.K.L. and D.R.S. organized and led the working group. D.R.S. carried out the simulation experiments. D.R.S. with the help from J.B.B. and W.K.L. analysed the data and wrote the manuscript. All authors contributed towards interpreting the data and improving the manuscript.

## Additional information

**Competing financial interests:** The authors declare no competing financial interests.

