## [Peer Review File · Nature Communications]

Reviewers' comments:

Reviewer #1 (Remarks to the Author):

This paper focuses specifically on temperate drylands and the effects of climate change on their extent and soil moisture dynamics. The questions are interesting and valuable and the work is extensive. At the same time, I have some questions regarding the interpretation of the results; the manuscript requires improved clarity to know whether the conclusions are supported by the results and to ensure that the paper will influence thinking in the field. I offer some specific questions and suggestions with respect to the major claims of the paper.

Net Loss of Temperate Drylands

Lines 105-119

The authors use a categorization based on temperature and aridity to determine temperate drylands under current conditions as well as under future climate scenarios. They find that temperate drylands decrease by a net of 27-29% (2.4×10^6 km² out of 8.3×10^6 km²; 36% of current TDL converted to something else; adding 9% more).

While those calculations seem straightforward, I found the statement on line 112-113 misleading: "This contraction of temperate drylands differs from predictions that drylands in total may increase by 5-23% globally.^{1,2}" I recommend that the authors find another way to say this, since most of the lost temperate drylands are converted to (sub)tropical drylands (by their own acknowledgement). Perhaps, "While other studies indicate that drylands in total may increase by 5-23% globally (1,2), that general statement masks our result that temperate drylands may contract while (sub)tropical drylands expand."

Increasing cold-season precipitation

To be honest, I do not see the data that support this claim. By my reading of Supplementary Table 2, annual rainfall increases under many combinations of regions and contraction/stable/expansion (western Mediterranean being the exception). However, seasonality appears to be decreasing compared with the current state for almost all combinations of region and contraction/stable/expansion. It may be that winter precipitation is indeed increasing (e.g., seasonality is decreasing but more total precipitation means more winter precipitation). I suggest that, if the authors wish to claim a result with respect to changes in winter precipitation, they calculate and show that quantity directly (i.e., compute winter precipitation and changes to it in addition to MAP).

Duration of Ecological Drought

Lines 144-145

I think it may be worth noting in the text that East Asia is the only region for which warm weather and precipitation are in-phase. This would provide some context for why East Asia seems to behave differently from the other regions under climate change.

Lines 146-148

In addition to stating the absolute changes in length of ecological drought, I suggest that the authors also articulate the percent change in drought length. Since ecological drought is longer in deep layers than shallow layers under current conditions, the difference in percent change may not be as significant.

Transpiration derived from deep layers (lines 150-159, Figure 3, and lines 454-457)

The authors claim that more transpiration will occur from shallow layers than deep layers under climate change. I suggest that they improve the clarity of what is meant by this. I could imagine a couple of interpretations:

- in absolute terms (mm/year), transpiration from surface layers is greater than transpiration from deep layers after climate change
- in relative terms, the importance of transpiration from surface layers increases under climate change; i.e., the fraction of total transpiration that derives from deep layers decreases under climate change (independent of whether the absolute of deep-layer transpiration amount goes up or down and independent of whether the amount of deep-layer transpiration is greater than or less than the amount from shallow layers).

I recommend being clear and specific about all of these potential interpretations. This seems especially important given the potentially self-contradictory claim that "future vegetation will extract more water from surface rather than deep layers" since "soil drying [i.e., extraction of water by plants] is more pronounced for deeper layers." What is the mechanism by which deep soils are drier but do not provide as much of the transpiration stream?

In Figure 3, it is unclear to me what the blue and orange colors represent - are they the absolute changes in the fraction of transpiration coming from deep layers (i.e., are they bounded by -green value to +(1-green value))? Or, are they the relative change in the proportion?

In lines 454-457, the authors state that the "mean annual proportion of transpiration derived from deep soil moisture" is "the ratio of transpiration... from shallow soil layers to transpiration... from deep soil layers." Obviously, something is amiss here and requires clarification. Should this be the ratio of deep transpiration to total?

Lines 153-155 indicate the decrease in the proportion of water transpired from deeper layers across the regions. While the proportion decreases (by amounts ranging from 2-11%), does the absolute amount of transpiration from the deep layers increase or decrease? It seems that, given the increase in ecological drought, the absolute transpiration would increase, while the proportion may decrease (since total transpiration increases). Clarity on this front would enhance the paper.

Relationship between change in proportion of deep transpiration and current value

Lines 159-163

The authors state that they find "a negative relationship between future change in the proportion of transpiration derived from deeper soil moisture and the current value (Fig. 4)"

The individual polynomial regressions (lines) in Figure 4 seem to support this claim (with the exception of perhaps North America) for changes within a region. However, if the median values are used to represent variation among regions, then the relationship seems to be one in which the value of change is smaller for larger values of the current state. Thus, while the figure supports the claim of homogenization of plant-water uptake within regions, I do not think it supports the claim of homogenization among regions. Also, the figure caption indicates "shaded areas", but I could not see any shaded areas; in my version, I see a series of traces and the median values for each region.

Overall, I found the figure captions to be limited and sometimes unclear. Improving clarity will increase the impact of the paper.

Also, line 128, schistosomiasis is misspelled.

Reviewer #2 (Remarks to the Author):

The manuscript, "Climate change reduces extent of temperate drylands and intensifies drought in deep soils" written by Daniel and coauthors predict a net loss of a third of current temperate dryland extent due to climate change in the late 21st century. Projections suggest the ecological droughts in deep soils (>20cm) will substantially increase and vegetation will be increasingly reliant on surface soil moisture, favoring shallow-rooted over deep-rooted vegetation. This result is meaningful and important for dryland woody encroachment research.

In general, it need to address those comments listed below before accepted this paper for publication.

1. The authors pointed out the reduction extent of temperate dryland under RCP4.5 and RCP8.5. However, does this phenomenon also exist in the historical warming period (1948-2015)? If this phenomenon is robust, the authors should give a figure showing the reductions of temperate dryland during 1948-2015 or longer historical records.
2. The definition of temperate dryland was given in methods. But how is the (sub)tropical dryland? Generally, subtropical and temperate dryland is determined by latitude, but in L114, why will the current temperate drylands be converted mainly to warmer subtropical drylands? In Fig. 1, do the contracting (orange) area indicate the conversion from temperate to subtropical dryland? If not, please give a distribution of this kind of conversion.
3. L150-152: the Fig. 3 and extended data fig. 6 show the proportion of transpiration derived from deep soil moisture (>20cm depth) under RCP8.5 and current climate. Only based on these two figures, the conclusion of "as a consequence of differential drying of deep vs. surface soil layers, future vegetation will extract more water from surface rather than deeper layers" can not be demonstrated because of lacking of the transpiration derived from the surface soil layer.
4. A mainline is missing to link the reduction of temperate dryland and increasing drying in deep soil. The reasons of reduction of temperate dryland and increasing drying in deep soil are both not explained.

Specific comments:

1. In Fig.2, how was the ecological drought determined? I do not find the definition in the methods.

Reviewer #3 (Remarks to the Author):

Review of "Climate change reduces extent of temperate drylands and intensifies drought in deep soils" by Schlaepfer et al. submitted to Nature Communications

Recommendation: Reject

General comments:

This article investigates the changes in ecological climate zones as a function of climate change by driving an ecohydrological model with CMIP5 simulations. It comes to the conclusion that the extent of drylands is decreased in future and that drought is intensified in deep soils.

While this article addresses an important topic, I can unfortunately not recommend its publication in Nature Communication and I evaluate that it would be better suitable for a specialist journal. The main concern is that the authors only use a single ecohydrological model for their assessment (despite the large author team). I am not familiar with the employed model ("SOILWAT"), but I assume that other ecohydrological models exist that could be used for comparison. More importantly, it is not clear that the model is suitable for one main application of the article, namely the assessment of changes in soil moisture content at various depths. In hydroclimatological research, extensive multi-model experiments have been conducted to assess both past and projected changes in land water characteristics. All of these investigations have shown a strong model dependency of results and large uncertainties in the representation of soil moisture in hydrological, land surface, and climate models (e.g. Dirmeyer et al. 2006, Orlowsky and Seneviratne 2013, Prudhomme et al. 2014, Mao et al. 2015). Hence, the study's results are rather anecdotal in the present form without a comparison with results from other land surface, ecohydrological, hydrological or ecosystem models.

References:

Dirmeyer, P.A. et al. 2006: GSWP-2 Multimodel Analysis and Implications for Our Perception of the Land Surface. Bull. Am. Met. Soc, DOI:10.1175/BAMS-87-10-138

Mao, J., et al. 2015: Disentangling climatic and anthropogenic controls on global terrestrial evapotranspiration trends. Environ. Res. Lett., 10, 094008

Orlowsky, B., and S.I. Seneviratne, 2013: Elusive drought: uncertainty in observed trends and short- and long-term CMIP5 projections. Hydrol. Earth Syst. Sci., 17, 1765-1781.

Prudhomme C., et al. 2014: Hydrological droughts in the 21st century, hotspots and uncertainties from a global multimodel ensemble experiment. PNAS, www.pnas.org/cgi/doi/10.1073/pnas.1222473110

Detailed responses to reviewer's comments

Net Loss of Temperate Dryland areas

Reviewer #1: Lines 105-119. The authors use a categorization based on temperature and aridity to determine temperate drylands under current conditions as well as under future climate scenarios. They find that temperate drylands decrease by a net of 27-29% (2.4×10^6 km² out of 8.3×10^6 km²; 36% of current TDL converted to something else; adding 9% more). While those calculations seem straightforward, I found the statement on line 112-113 misleading: "This contraction of temperate drylands differs from predictions that drylands in total may increase by 5-23% globally.^{1,2}" I recommend that the authors find another way to say this, since most of the lost temperate drylands are converted to (sub)tropical drylands (by their own acknowledgement). Perhaps, "While other studies indicate that drylands in total may increase by 5-23% globally (1,2), that general statement masks our result that temperate drylands may contract while (sub)tropical drylands expand."

→ We agree and incorporated this good suggestion for clarification.

Reviewer #2: 1. The authors pointed out the reduction extent of temperate dryland under RCP4.5 and RCP8.5. However, does this phenomenon also exist in the historical warming period (1948-2015)? If this phenomenon is robust, the authors should give a figure showing the reductions of temperate dryland during 1948-2015 or longer historical records.

→ This is an excellent suggestion by the reviewer to consider areal changes in temperate drylands through the 20th century. However, other authors have addressed this question and estimated contemporary increase of aridity, i.e., expanding global drylands, during the second half of the 20th century. Feng et al. (2013) suggest a 4% increase during the last 50 years and Dai's (2012) analysis suggests an 8% increase. Despite observations of expanding aridity (i.e., climatic norm), attribution of changes in drought (i.e., deviations from normal) to human activity for the second half of the 20th century remains uncertain (IPCC 2014; Sherwood *et al.* 2014; Trenberth *et al.* 2014).
→ We investigated the extent of temperate dryland for 2070-2100 against conditions for 1980-2010, We do not have the appropriate data going back to 1948 and doing the calculations for all of our grid cells would be a project in and of itself and beyond our current project and manuscript objectives. We believe that a re-analysis by our project for the 20th century would not contribute considerably new insights to what others have published; instead, we base our insights on their result and added their estimates of 20th century change to the introduction section for clarification.

Reviewer #2: 2. The definition of temperate dryland was given in methods. But how is the (sub)tropical dryland? Generally, subtropical and temperate dryland is determined by latitude, but in L114, why will the current temperate drylands be converted mainly to warmer subtropical drylands? In Fig. 1, do the contracting (orange) area indicate the conversion from temperate to subtropical dryland? If not, please give a distribution of this kind of conversion.

→ Our definitions of temperate versus tropical were based on climate, rather than latitude, and we derived these definitions from the commonly used Trewartha classification scheme. Thus, when climate changes, the extent of tropical, subtropical

and temperate regions can change accordingly.

→ We thank the reviewer for pointing out that our description was not clear enough. We improved our description of dryland and temperateness and added a short explanation at the beginning of the result section and improved the caption of Fig. 1. The orange areas in Fig. 1 are locations that are currently temperate and dryland, but not expected to meet one of those two criteria in the future. Thus, they may have converted from temperate to subtropical, or from dryland to non-dryland (a complete table of conversions (i.e., attributions to the defining factors) can be found in the Supplementary Table 3).

Increasing cold-season precipitation

Reviewer #1: To be honest, I do not see the data that support this claim. By my reading of Supplementary Table 2, annual rainfall increases under many combinations of regions and contraction/stable/expansion (western Mediterranean being the exception). However, seasonality appears to be decreasing compared with the current state for almost all combinations of region and contraction/stable/expansion. It may be that winter precipitation is indeed increasing (e.g., seasonality is decreasing but more total precipitation means more winter precipitation). I suggest that, if the authors wish to claim a result with respect to changes in winter precipitation, they calculate and show that quantity directly (i.e., compute winter precipitation and changes to it in addition to MAP).

→ We recognize that we have used the term '(precipitation) seasonality' too loosely. 'Seasonality' can be understood in different ways, e.g., statistically as a component of time-series analysis of temperature data (Pezzulli et al. 2005); astronomically by solstices and equinoxes or by calendar months (Trenberth 1983); as the standard deviation of mean monthly temperature or as the coefficient of variation of mean monthly precipitation (BioClim/WorldClim, <http://www.worldclim.org/bioclim>, Nix 1986); as the inverse of mean annual temperature (Xu et al. 2013); etc. The definition of (precipitation) seasonality used by our manuscript is based on work by Sala and colleagues (1997) who define the 'overlap of the wet and warm season' as a function of "the mean annual Pearson correlation coefficient between mean monthly temperature (°C) and monthly precipitation (mm). +1 indicates a perfect match between the warm and wet season; -1 indicates a perfect match between the cold and the wet season" (Supplementary Table 6).

A decrease of this correlation describes a shift of precipitation from the warm to the cold season given no or a change in temperature which is uniform across months; that means, for constant or increased MAP, there is an increase in cold-season precipitation. We contend thus that our data support our claim of an increase in cold-season precipitation.

→ To address this comment, (i) we changed the term 'precipitation seasonality' to 'warm/wet-season overlap' and added our definition to the method sections, and (ii) we added winter precipitation (defined meteorologically as precipitation during December-February on the northern hemisphere and during June-August on the southern hemisphere, Trenberth 1983) to Supplementary Table 2 and as map in a new Supplementary Figure 5. We found increases in winter precipitation for our regions

North America, East Asia, Western & Central Asia, and the southern portion of South America. Most of the area of the Mediterranean Basin was predicted to experience decreases in winter precipitation.

Duration of Ecological Drought

Reviewer #1: Lines 144-145. I think it may be worth noting in the text that East Asia is the only region for which warm weather and precipitation are in-phase. This would provide some context for why East Asia seems to behave differently from the other regions under climate change.

→ We incorporated this good suggestion.

Reviewer #1: Lines 146-148. In addition to stating the absolute changes in length of ecological drought, I suggest that the authors also articulate the percent change in drought length. Since ecological drought is longer in deep layers than shallow layers under current conditions, the difference in percent change may not be as significant.

→ We calculated the percent change in drought length as suggested and found the same pattern, i.e., ecological drought in deep soil layers is becoming relatively and absolutely longer and this change is not detectable/more varied in the shallow layers. We added the percent change to the manuscript text.

Reviewer #2: 1. In Fig.2, how was the ecological drought determined? I do not find the definition in the methods.

→ We agree that our definition of ecological drought was in a hard-to-find location in the manuscript. To address this comment, we added an abbreviated version of our definition to the caption of Figure 2. We also expanded the introduction section by highlighting the different types of drought definitions (meteorological, etc.) and introduce a standard definition of ecological drought (“prolonged and widespread deficit in naturally available water supplies [...] that create multiple stresses across ecosystems”; U.S. Geological Survey, U.S. Climate Science Centers, and the Science for Nature and People Partnership) which for instance can be found here <https://nccwsc.usgs.gov/science/ecological-drought> and <http://snappartnership.net/groups/ecological-drought/> (accessed July 9, 2016).

→ Our definition of ecological drought, which implements the standard definition and in a measurable form, can be found in the second paragraph of the subsection 'Analysis of response variables' of the methods: "We estimated the mean annual duration of continuous ecological droughts during growing periods for surface soil layers of 0-20 cm depth (DDGP0) and for deep soil layers > 20 cm depth (DDGP20) as the longest snow-free, frost-free period when soil water potential (SWP) < -3.0 MPa continuously."

Transpiration derived from deep layers

Reviewer #1: (lines 150-159, Figure 3, and lines 454-457) The authors claim that more transpiration will occur from shallow layers than deep layers under climate change. I suggest that they improve the clarity of what is meant by this. I could imagine a couple of interpretations:

- in absolute terms (mm/year), transpiration from surface layers is greater than transpiration from

deep layers after climate change

- in relative terms, the importance of transpiration from surface layers increases under climate change; i.e., the fraction of total transpiration that derives from deep layers decreases under climate change (independent of whether the absolute of deep-layer transpiration amount goes up or down and independent of whether the amount of deep-layer transpiration is greater than or less than the amount from shallow layers).

→ The reviewer is correct that we provided not sufficient data to support our claims and that our writing was ambiguous. We produced additional maps and table sections for transpiration derived from soil moisture in shallow soil layers and for transpiration derived from deep layers (Supplementary Figs 7–8 and in Supplementary Tables 4–5). We incorporated these more detailed results in the text (see result subsection ‘Duration and distribution of ecological droughts under climate change’) and clarified our statements. It turns out that both interpretations by the reviewer are correct, i.e., that for most regions transpiration derived from shallow layers is increasing in absolute and relative terms.

Reviewer #1: I recommend being clear and specific about all of these potential interpretations. This seems especially important given the potentially self-contradictory claim that "future vegetation will extract more water from surface rather than deep layers" since "soil drying [i.e., extraction of water by plants] is more pronounced for deeper layers." What is the mechanism by which deep soils are drier but do not provide as much of the transpiration stream?

→ This was a misunderstanding due to unclear writing on our parts. We were referring to increasing ecological droughts due to climate change when we wrote ‘soil drying’ – we did not mean extraction of water by plants. Our results are coherent in that deep layers will experience longer periods with low soil moisture and plants will extract a smaller proportion of their total water extraction from deep layers. We have re-worded the text to avoid such ambiguity (see result subsection ‘Duration and distribution of ecological droughts under climate change’).

Reviewer #1: In Figure 3, it is unclear to me what the blue and orange colors represent - are they the absolute changes in the fraction of transpiration coming from deep layers (i.e., are they bounded by -green value to +(1-green value))? Or, are they the relative change in the proportion?

→ The blue-orange color gradient in panels b-f of Fig. 3 indicate the absolute change in the proportion of transpiration from deep layers between future and current values. We have re-worded the figure caption for clarity.

Reviewer #1: In lines 454-457, the authors state that the "mean annual proportion of transpiration derived from deep soil moisture" is "the ratio of transpiration... from shallow soil layers to transpiration... from deep soil layers." Obviously, something is amiss here and requires clarification. Should this be the ratio of deep transpiration to total?

→ Yes, this was a typo and the ratio is deep to total. We have fixed this error.

Reviewer #1: Lines 153-155 indicate the decrease in the proportion of water transpired from deeper layers across the regions. While the proportion decreases (by amounts ranging from 2-11%), does the absolute amount of transpiration from the deep layers increase or decrease? It

seems that, given the increase in ecological drought, the absolute transpiration would increase, while the proportion may decrease (since total transpiration increases). Clarity on this front would enhance the paper.

→ We produced additional maps and table sections for transpiration derived from soil moisture in shallow soil layers and for transpiration derived from deep layers (Supplementary Figs 7–8 and in Supplementary Tables 4–5). Please see our detailed response to the first comment in this section of comments on ‘Transpiration derived from deep layers’.

Reviewer #2: 3. L150-152: the Fig. 3 and extended data fig. 6 show the proportion of transpiration derived from deep soil moisture (>20cm depth) under RCP8.5 and current climate. Only based on these two figures, the conclusion of "as a consequence of differential drying of deep vs. surface soil layers, future vegetation will extract more water from surface rather than deeper layers" can not be demonstrated because of lacking of the transpiration derived from the surface soil layer.

→ We agree with the reviewer. We produced additional maps and table sections for transpiration derived from soil moisture in shallow soil layers and for transpiration derived from deep layers (Supplementary Figs 7–8 and in Supplementary Tables 4–5). Please see our detailed response to the first comment in this section of comments on ‘Transpiration derived from deep layers’.

Reviewer #2: 4. A mainline is missing to link the reduction of temperate dryland and increasing drying in deep soil. The reasons of reduction of temperate dryland and increasing drying in deep soil are both not explained.

→ The reviewer is correct that change in temperate dryland area and changes in deep soil moisture are somewhat related because they do both respond to changes in climatic conditions. However, these two responses are not causally linked in our analysis – e.g. they both response to climate but don’t affect each other. Changes in the distribution of temperate drylands are calculated directly from changes in climatic conditions, specifically three climate variables, and Supplementary Table 3 provides detailed quantification of the contribution of each variable to contraction and expansion of temperate drylands. Change in deep soil moisture is an ecohydrological variable that is estimated by our soil water balance model. We have clarified the relationships among these variables in the result section.

Relationship between change in proportion of deep transpiration and current value

Reviewer #1: Lines 159-163. The authors state that they find "a negative relationship between future change in the proportion of transpiration derived from deeper soil moisture and the current value (Fig. 4)". The individual polynomial regressions (lines) in Figure 4 seem to support this claim (with the exception of perhaps North America) for changes within a region. However, if the median values are used to represent variation among regions, then the relationship seems to be one in which the value of change is smaller for larger values of the current state. Thus, while the figure supports the claim of homogenization of plant-water uptake within regions, I do not think it supports the claim of homogenization among regions.

→ This is a very good point. Our inferences related to the potential consequences of the

among-region “global” negative relationship between decrease in proportion of deep transpiration and current proportion deep transpiration, while we did not interpret the within-region relationships. As the reviewer points out, two of the regions (North America and West Asia) do not have clear negative relationships, so the implications for homogenization do not apply within those regions. We have modified the text in the final paragraph of the results section to explicitly identify that the negative relationship (and associated implications for homogenization) exists among regions and only within some of the regions, but not all.

Reviewer #1: Also, the figure caption [of Fig. 4] indicates "shaded areas", but I could not see any shaded areas; in my version, I see a series of traces and the median values for each region.

→ We are sorry that the pdf version of our figures did not show all the information for this reviewer. We have redrawn the shading of the figures and saved them in a different format. We will double-check that the PDF validation during submission will show the shaded areas.

Sensitivity of land surface/soil moisture results to different models

Reviewer #3: While this article addresses an important topic, I can unfortunately not recommend its publication in Nature Communication and I evaluate that it would be better suitable for a specialist journal. The main concern is that the authors only use a single ecohydrological model for their assessment (despite the large author team). I am not familiar with the employed model ("SOILWAT"), but I assume that other ecohydrological models exist that could be used for comparison. More importantly, it is not clear that the model is suitable for one main application of the article, namely the assessment of changes in soil moisture content at various depths.

→ The reviewer is correct that other models exist, but the implication that they could be run with the same set of input data is not correct. The models vary widely in the amount of detail they include for each of the key processes and in the temporal scales for which they are intended to run. Soilwat is well documented and tested for this scale of application. We think our previous work, which has been published in the refereed literature establishes that SOILWAT is suitable for assessing changes in soil water content at multiple soil depths (Lauenroth et al. 2006; Schlaepfer et al. 2012b; Schlaepfer et al. 2012a; Bradford et al. 2014; Lauenroth et al. 2014; Schlaepfer et al. 2014; Palmquist et al. 2016). No similar models are available that could be run using our data set.

Reviewer #3: In hydroclimatological research, extensive multi-model experiments have been conducted to assess both past and projected changes in land water characteristics. All of these investigations have shown a strong model dependency of results and large uncertainties in the representation of soil moisture in hydrological, land surface, and climate models (e.g. Dirmeyer et al. 2006, Orlowsky and Seneviratne 2013, Prudhomme et al. 2014, Mao et al. 2015). Hence, the study's results are rather anecdotal in the present form without a comparison with results from other land surface, ecohydrological, hydrological or ecosystem models.

→ We agree that ideally we would have captured uncertainty in representation of ecohydrological processes by using multiple ecohydrological models. This was

unfortunately not possible because no similar models exist (see previous response).

→ We have nevertheless estimated several important sources of uncertainty by forcing our ecohydrological model with input from two climate scenarios (results for RCP8.5 in the manuscript, those for RCP4.5 in the Supplementary Materials) and 16 global circulation models (GCMs). For instance, we highlight the variation among the outcome from the 16 GCMs in Fig. 1 where the insets capture the agreement/disagreement among GCMs in areal shifts and in Fig. 2 where we account for the variation among GCMs in the estimation of ecological drought. All our other results in the text and in the figures capture the range of variation derived from simulating soil moisture related response variables when run with climate predictions from the 16 GCMs.

→ Many recent publications in high-profile peer-reviewed journals, including Nature Communications, present results from a single model. For instance, Nature Communications published research based on a single global circulation model from Yoon et al. (2015) and Thieblemont et al. (2015). A single ecosystem/hydrology model underlies the results by Carroll et al. (2015) and a single soil model the work by (Sattari et al. 2016). Dong et al. (2016) used a single weather model for their sensitivity analysis and Mellin et al. (2016) one statistical model (out of many possible) for a 'species distribution' study. Thus, our study, which covers 32 different climate outcomes, goes well beyond what most other recent papers in Nature Communications and other similar high profile journals offer.

→ The reviewer is correct in pointing out that GCMs incorporate a representation of soil moisture (albeit at a very coarse resolution). Our simulation effort addresses exactly this gap in our knowledge by using an ecohydrological model that is capable of simulating the soil profile and relevant ecohydrological processes at a much higher resolution (see previous response). We have added a new comparison between GCM soil moisture data, where available, with results from our simulations (see Supplementary Fig. 10 and Supplementary Table 10). The comparison is favorable with an overall agreement level for the historical time period of 0.89 ± 0.07 (mean \pm SD among 7 GCM-SOILWAT comparisons) as well as for the future time period under RCP8.5 of 0.92 ± 0.04 . Regional agreement is mostly similarly high. GCM-SOILWAT agreement, however, was low for Eastern Asia in the historic time period with 0.37 ± 0.21 , which increased to 0.67 ± 0.09 for the future period under the RCP8.5 scenario. Our simulations for the historic time period were run with observation-based weather data, whereas the GCM output represents hindcasts. For the future time periods, the representation of climate conditions for our simulations were based on GCM output. Thus, we would expect a higher agreement between our simulation results and those from GCMs for the future time periods than for the historic period. Freedman et al. (2014) compared GRACE satellite observations of terrestrial water storage with GCM predictions for 2003-2012 for the Mississippi River Basin and found good agreement in overall aggregated values, but considerable GCM deviations spatially and in water flux partitioning. In a similar exercise, Wu et al. (Wu et al. 2015) compared GRACE data to GCM predictions to select GCMs for a hydrological impact assessment and found noticeable variation among GCM soil moisture predictions including GCMs with cycles that do not match the seasonal variation. It is not surprising that we found some deviations between SOILWAT and GCM soil moisture values as well, particularly across Eastern Asia, which is a region where several GCMs demonstrate difficulties in representing the monsoonal precipitation

regime (Park *et al.* 2016).

General comments:

Reviewer #1: Overall, I found the figure captions to be limited and sometimes unclear.

Improving clarity will increase the impact of the paper.

→ We have re-worded the figure captions for clarity. In particular, we expanded the caption for figure 1 to more comprehensively describe what the inset histograms depict. We also have re-colored Fig. 1 to improve interpretability of the color gradients.

Reviewer #1: Also, line 128, schistosomiasis is misspelled.

→ Corrected

References

- Bradford JB, Schlaepfer DR, Lauenroth WK (2014) Ecohydrology of adjacent sagebrush and lodgepole pine ecosystems: The consequences of climate change and disturbance. *Ecosystems*, **17**, 590-605.
- Carroll MJ, Heinemeyer A, Pearce-Higgins JW *et al.* (2015) Hydrologically driven ecosystem processes determine the distribution and persistence of ecosystem-specialist predators under climate change. *Nature Communications*, **6**, 7851.
- Dai A (2012) Increasing drought under global warming in observations and models. *Nature Climate Change*, **3**, 52-58.
- Dong W, Lin Y, Wright JS *et al.* (2016) Summer rainfall over the southwestern Tibetan Plateau controlled by deep convection over the Indian subcontinent. *Nature Communications*, **7**, 10925.
- Feng S, Fu Q (2013) Expansion of global drylands under a warming climate. *Atmospheric Chemistry and Physics*, **13**, 10081-10094.
- Freedman FR, Pitts KL, Bridger AFC (2014) Evaluation of CMIP climate model hydrological output for the Mississippi River Basin using GRACE satellite observations. *Journal of Hydrology*, **519**, Part D, 3566-3577.
- IPCC (2014) (eds Stocker TF, Qin D, Plattner G-K *et al.*), pp. 1535. Cambridge University Press, Cambridge, United Kingdom and New York, NY, USA.
- Lauenroth WK, Bradford JB (2006) Ecohydrology and the partitioning AET between transpiration and evaporation in a semiarid steppe. *Ecosystems*, **9**, 756-767.
- Lauenroth WK, Schlaepfer DR, Bradford JB (2014) Ecohydrology of dry regions: storage versus pulse soil water dynamics. *Ecosystems*, **17**, 1469-1479.
- Mellin C, Mouillot D, Kulbicki M *et al.* (2016) Humans and seasonal climate variability threaten large-bodied coral reef fish with small ranges. *Nature Communications*, **7**, 10491.
- Nix H (1986) A biogeographic analysis of Australian elapid snakes. In *Atlas of Elapid snakes of Australia. Australian Flora and Fauna Series 7* (ed Longmore R). Australian Government Publishing Service, Canberra.

- Palmquist K, Schlaepfer DR, Lauenroth WK, Bradford JB (2016) Mid-latitude shrub steppe plant communities: Climate change consequences for soil water resources. *Ecology*, na/na.
- Park C, Min S-K, Lee D *et al.* (2016) Evaluation of multiple regional climate models for summer climate extremes over East Asia. *Climate Dynamics*, **46**, 2469-2486.
- Pezzulli S, Stephenson DB, Hannachi A (2005) The Variability of Seasonality. *Journal of Climate*, **18**, 71-88.
- Sala OE, Lauenroth WK, Golluscio RA (1997) Plant functional types in temperate semi-arid regions. In *Plant functional types: their relevance to ecosystem properties and global change* (eds Smith TM, Shugart HH, Woodward FI), pp. 217-233. Cambridge University Press, Cambridge.
- Sattari SZ, Bouwman AF, Martinez Rodriguez R, Beusen AH, van Ittersum MK (2016) Negative global phosphorus budgets challenge sustainable intensification of grasslands. *Nature Communications*, **7**, 10696.
- Schlaepfer DR, Lauenroth WK, Bradford JB (2012a) Consequences of declining snow accumulation for water balance of mid-latitude dry regions. *Global Change Biology*, **18**, 1988-1997.
- Schlaepfer DR, Lauenroth WK, Bradford JB (2012b) Ecohydrological niche of sagebrush ecosystems. *Ecohydrology*, **5**, 453-466.
- Schlaepfer DR, Lauenroth WK, Bradford JB (2014) Modeling regeneration responses of big sagebrush (*Artemisia tridentata*) to abiotic conditions. *Ecological Modelling*, **286**, 66-77.
- Sherwood S, Fu Q (2014) A drier future? *Science*, **343**, 737-739.
- Thieblemont R, Matthes K, Omrani NE, Kodera K, Hansen F (2015) Solar forcing synchronizes decadal North Atlantic climate variability. *Nature Communications*, **6**, 8268.
- Trenberth KE (1983) What are the Seasons? *Bulletin of the American Meteorological Society*, **64**, 1276-1282.
- Trenberth KE, Dai A, van der Schrier G, Jones PD, Barichivich J, Briffa KR, Sheffield J (2014) Global warming and changes in drought. *Nature Climate Change*, **4**, 17-22.
- Wu W-Y, Lan C-W, Lo M-H, Reager JT, Famiglietti JS (2015) Increases in the annual range of soil water storage at northern middle and high latitudes under global warming. *Geophysical Research Letters*, **42**, 3903-3910.
- Xu L, Myneni RB, Chapin Iii FS *et al.* (2013) Temperature and vegetation seasonality diminishment over northern lands. *Nature Climate Change*, **3**, 581-586.
- Yoon JH, Wang SY, Gillies RR, Kravitz B, Hipps L, Rasch PJ (2015) Increasing water cycle extremes in California and in relation to ENSO cycle under global warming. *Nature Communications*, **6**, 8657.

Reviewers' comments:

Reviewer #1 (Remarks to the Author):

This paper focuses specifically on temperate drylands and the effects of climate change on their extent and soil moisture dynamics. The questions are interesting and valuable and the work is extensive. The original manuscript suffered from some issues of clarity and interpretation of the results, and the authors have addressed most of those in their revised manuscript.

I have just one recommendation regarding the presentation of results regarding transpiration from deep and shallow layers. On lines 172-174, the authors articulate "Our simulations suggest that the importance of transpiration from shallow layers increases under climate change, i.e., the proportion of total transpiration that derives from deep layers decreases (Fig. 3)." This statement is well supported by Figure 3 and Supplemental Tables 4 and 5. I recommend concluding this statement with a period.

The next phrase, I found confusing: "...and additionally, that transpiration from shallow layers will increase more than transpiration from deep layers (Supplementary Figs 7-8)." For me, it is hard to directly compare the color maps between Figures S7-S8. When I look at Tables S4-S5, however, those data indicate to me that sometimes transpiration increases in the shallow layers and sometimes it decreases (e.g., T0 increases in South America for the areas under contraction, while it decreases in the Western Mediterranean), and the same is true for transpiration from the deep layers (e.g., T20 increases in East Asia and North America under contraction, while it decreases for South America and Central Asia). Thus, it seems confusing to state that "transpiration from shallow layers will increase more than transpiration from deep layers," which, to me, implies that transpiration always increases in both layers. I might suggest something like, "While transpiration from shallow and deep layers may increase or decrease under future conditions and across regions, the proportion of transpiration from deep layers shows a robust decline across all scenarios."

One last minor point - The color scheme used in Figure 2 and Figure 4 is presented in the caption for Figure 4 but not Figure 2. I recommend including it in both locations.

Reviewer #2 (Remarks to the Author):

The revised manuscript appropriately addressed all my comments/suggestions and significantly improved the manuscript. I recommend that the paper be accepted as it is.

Reviewer #3 (Remarks to the Author):

Review of Schlaepfer et al., submitted to Nature Communications (NCOMMS-16-00671-T)

Overall recommendation: Reject

General comments: The authors did efforts to improve the article in response to the reviewers' comments. However, I do still have major concerns which prevent me to recommend this article

for publication in Nature Communications.

My main concerns are related to the inferred aridity changes and their possible implications for ecosystems (the core of the article...).

There is indeed evidence that some regions are displaying increases in "aridity" measured as the ratio of potential evaporation (Epot) and precipitation under increased greenhouse gas forcing, however, there is no evidence that this would lead to stronger ecosystem moisture stress in drylands, as assumed by the authors (and apparently simulated by their model). The main issues are as follows:

* The study overlooks possible effects of increased CO₂ concentrations for the ecosystems, which could decrease their water stress even under increased Epot/P. The reason for this is that increase atmospheric CO₂ can increase the water-use efficiency of plants. The possible impact of this response has been discussed in recent publications (Prudhomme et al. 2014, PNAS; Roderick et al. 2015, WRR; Milly and Dunne 2016, Nature Climate Change). It could possibly strongly change the hydrological response of ecosystems. The authors should at least mention this issue and the possible implications of overlooking this response. This could be of less importance for a study focusing e.g. on hydrological changes, but for a study assessing possible changes in drought-related ecosystem response, this is in my view just not suitable for publication (even in another journal than Nature Communications).

* Even when not considering the CO₂ effects on water use efficiency, the aridity changes will have least impacts on the water balance of arid regions, because evapotranspiration is soil moisture limited there and thus increased demand (from Epot) does not imply increased actual evapotranspiration (see e.g. on this topic Gudmundsson et al. 2016, GRL). Hence increased aridity is mostly of interest in presently humid regions, which is not a topic of the present article.

* Finally, there are several recent publications questioning altogether the use of Epot as driver of evapotranspiration (and thus ecosystem response) in a warming climate, because it increases very strongly with increasing temperature and decreasing air humidity, but thereby reflects a response to the surface fluxes rather than a change as a driver of actual evapotranspiration (see also two points above). For instance, Roderick et al. (2015) show that net radiation increases much less than potential evaporation in a warming climate, and ultimately it is the available energy that constrains the surface water fluxes, not Epot itself (Epot is merely a construct to represent this effect). Milly et al. (2016) also provide further discussions on this topic.

Based on these issues, and in particular because the authors appear unaware of them, I can unfortunately not recommend publication of this article.

References:

Gudmundsson, L., P. Greve, and S.I. Seneviratne, 2016: The sensitivity of water availability to changes in the aridity index and other factors-A probabilistic analysis in the Budyko space. *Geophys. Res. Lett.*, DOI: 10.1002/2016GL069763.

Milly, P.C.D. and Dunne, K.A., 2016: Potential evapotranspiration and continental drying. *Nature Climate Change*, DOI: 10.1038/NCLIMATE3046

Prudhomme, C., et al., 2014: Hydrological droughts in the 21st century, hotspots and uncertainties from a global multimodal ensemble experiment. *Proc. Natl. Acad. Sci.*, www.pnas.org/cgi/doi/10.1073/pnas.1222473110

Roderick, M.L., P. Greve, and G.D. Farquhar, 2015: On the assessment of aridity with changes in

atmospheric CO2. Water Resources Research, DOI: 10.1002/2015WR017031

Detailed responses to reviewer's comments

Reviewer #1:

I have just one recommendation regarding the presentation of results regarding transpiration from deep and shallow layers. On lines 172-174, the authors articulate, "Our simulations suggest that the importance of transpiration from shallow layers increases under climate change, i.e., the proportion of total transpiration that derives from deep layers decreases (Fig. 3)." This statement is well supported by Figure 3 and Supplemental Tables 4 and 5. I recommend concluding this statement with a period.

→ We incorporated this useful suggestion and split the sentence in two.

The next phrase, I found confusing: "...and additionally, that transpiration from shallow layers will increase more than transpiration from deep layers (Supplementary Figs 7-8)." For me, it is hard to directly compare the color maps between Figures S7-S8. When I look at Tables S4-S5, however, those data indicate to me that sometimes transpiration increases in the shallow layers and sometimes it decreases (e.g., T0 increases in South America for the areas under contraction, while it decreases in the Western Mediterranean), and the same is true for transpiration from the deep layers (e.g., T20 increases in East Asia and North America under contraction, while it decreases for South America and Central Asia). Thus, it seems confusing to state that "transpiration from shallow layers will increase more than transpiration from deep layers," which, to me, implies that transpiration always increases in both layers. I might suggest something like, "While transpiration from shallow and deep layers may increase or decrease under future conditions and across regions, the proportion of transpiration from deep layers shows a robust decline across all scenarios."

→ The reviewer is correct that this phrase was an oversimplification of our results. Also, the order of our statements was confusing: 1) global trend in T0/T (lines 172-174); 2) global trend T0 vs. T20 (lines 174-175); 3) regional trends in T0/T (lines 176-180); 4) regional trends in T0 vs. T20 (lines 180-185). The 4th section addressed the geographic heterogeneity of increases and decreases in T0 and T20 (lines 180-185), which the reviewer justifiably highlights in this comment.

→ To address this comment, we re-organized the structure of this paragraph to: i) trends in T0/T with (i-1) global and (i-2) regional results; ii) trends in T0 vs. T20 with (i-1) global and (i-2) regional results providing details of the variable geographic pattern of both increases and decreases. We also deleted the phrase in question.

One last minor point - The color scheme used in Figure 2 and Figure 4 is presented in the caption for Figure 4 but not Figure 2. I recommend including it in both locations.

→ This is a good point that improves readability. We expanded the caption of Fig. 2 with an explanation of the color code.

Reviewer #2:

The revised manuscript appropriately addressed all my comments/suggestions and significantly

improved the manuscript. I recommend that the paper be accepted as it is.
→ We appreciate the reviewer's positive evaluation.

Reviewer #3:

Review of Schlaepfer et al., submitted to Nature Communications (NCOMMS-16-00671-T)

Overall recommendation: Reject

General comments: The authors did efforts to improve the article in response to the reviewers' comments. However, I do still have major concerns which prevent me to recommend this article for publication in Nature Communications.

My main concerns are related to the inferred aridity changes and their possible implications for ecosystems (the core of the article...).

There is indeed evidence that some regions are displaying increases in "aridity" measured as the ratio of potential evaporation (Epot) and precipitation under increased greenhouse gas forcing, however, there is no evidence that this would lead to stronger ecosystem moisture stress in drylands, as assumed by the authors (and apparently simulated by their model). The main issues are as follows:

→ Our manuscript took two perspectives: (1) a climatological viewpoint on aridity by delineating temperate dryland regions with an aridity index for historical and future time periods (similar to approaches taken, e.g., by Feng *et al.* 2013; Huang *et al.* 2015) and (2) an ecohydrological approach on ecological drought by executing a simulation experiment with a process-based daily ecosystem water balance model (Lauenroth *et al.* 2006; Schlaepfer *et al.* 2012; Bradford *et al.* 2014; Palmquist *et al.* 2016) to infer historical and future ecosystem moisture stress patterns in drylands. It is the nature of investigations that address future time periods that evidence to support forecasts of future conditions cannot exist. However, our model simulations agree well on an aggregated time-scale with the coarser representation of soil moisture of Earth System Models for future time periods. We argue that these favorable comparisons support the results and conclusions on climatological aridity and on ecological drought (soil moisture drought) presented in our manuscript.

* The study overlooks possible effects of increased CO₂ concentrations for the ecosystems, which could decrease their water stress even under increased Epot/P. The reason for this is that increase atmospheric CO₂ can increase the water-use efficiency of plants. The possible impact of this response has been discussed in recent publications (Prudhomme et al. 2014, PNAS; Roderick et al. 2015, WRR; Milly and Dunne 2016, Nature Climate Change). It could possibly strongly change the hydrological response of ecosystems. The authors should at least mention this issue and the possible implications of overlooking this response. This could be of less importance for a study focusing e.g. on hydrological changes, but for a study assessing possible changes in drought-related ecosystem response, this is in my view just not suitable for publication (even in another journal than Nature Communications).

→ We incorporated the reviewer's suggestion and added a description in the revised manuscript that our simulation experiment assumed that ambient CO₂-concentration has a negligible net effect on modeled vegetation. Thus, our model could have led to an overestimate of soil drying (but some studies find a zero or even negative net effect of elevated CO₂ on soil moisture in dry ecosystems, e.g., see Schymanski *et al.* 2015; Mueller *et al.* 2016). We also want to point out that even though the physiological response of intrinsic water-use efficiency and photosynthesis at the leaf scale to CO₂-concentration are well understood (e.g., Bagley *et al.* 2015), the net impacts on ecosystem-scale transpiration and soil moisture depend on how enhanced leaf-level water use efficiency trade-off against increased foliage biomass over longer time periods, and the outcomes are certainly not yet clear (e.g., as demonstrated by FACE experiments, such as the one in the semiarid northern-mixed prairie Morgan *et al.* 2004; Morgan *et al.* 2011; Mueller *et al.* 2016). The long-term outcome also depends strongly on how CO₂-effects unfold in the context of nutrient vs. water limitation (Reich *et al.* 2014; reviewed by Norby *et al.* 2016) and energy vs. water limitation (Schymanski *et al.* 2015). Thus, increases in intrinsic leaf-level water-use efficiency may lead to a positive response in biomass and transpiration in water-limited systems (as opposed to energy-limited systems) and potentially to a decrease in soil moisture over the long run (Schymanski *et al.* 2015; Mueller *et al.* 2016). These and several additional issues remain to be addressed (e.g., as reviewed by Norby *et al.* 2011; Smith *et al.* 2013; Pugh *et al.* 2016). The importance of these uncertainties is illustrated by the large range of reported values for CO₂-responses of ecosystem-scale water-use efficiency (observations/experimental values range from 0% to +120%), transpiration (-14% to +11%), productivity (0% to 40%), and soil moisture (-20% to +10%). All of these interacting effects need to be appropriately represented in ecosystem/vegetation/Earth simulation models before they will be able to represent the full range of experimental and historical observations (e.g., De Kauwe *et al.* 2013; Walker *et al.* 2014; Zaehle *et al.* 2014; Frank *et al.* 2015; Huang *et al.* 2016). Our results, which assumed a negligible net effect of CO₂-concentration on ecosystem-scale transpiration and soil moisture, are well within the range of observations and models. Current Earth system models disagree so strongly over the net outcome in response to future CO₂-concentration that the earth system science communities' understanding of these effects has been described as "very low" (Medlyn *et al.* 2015).

* Even when not considering the CO₂ effects on water use efficiency, the aridity changes will have least impacts on the water balance of arid regions, because evapotranspiration is soil moisture limited there and thus increased demand (from Epot) does not imply increased actual evapotranspiration (see e.g. on this topic Gudmundsson *et al.* 2016, GRL). Hence increased aridity is mostly of interest in presently humid regions, which is not a topic of the present article.

→ We agree with the reviewer that actual evapotranspiration in dry regions is limited strongly by soil moisture and not by potential evapotranspiration rates *per se* (see previous work, e.g., Lauenroth *et al.* 2006; Schlaepfer *et al.* 2014). The reviewer may have misunderstood our study approach: First, we used an aridity index (Epot / P; an index which is widely used in the literature) only for the purpose of climatologically identifying our study regions because it is easily recognized by the broader scientific

community. Second, we applied a process-based ecohydrological daily water balance model to simulate every component of the water balance of the temperate dryland regions, identified in step 1, including actual evaporation, actual transpiration, deep drainage, soil moisture changes in many soil layers to infer patterns of ecological drought. That is, the aridity index did not drive our results on soil moisture—in agreement with the findings by Gudmundsson et al. (2016). SOILWAT is particularly well suited to simulate actual evapotranspiration in dry regions because the model accounts for limiting conditions by soil moisture at different depths, by accessibility of moisture depending on soil texture and plant rooting characteristics, and by canopy and litter shading (Lauenroth *et al.* 2006; Bradford *et al.* 2014).

* Finally, there are several recent publications questioning altogether the use of Epot as driver of evapotranspiration (and thus ecosystem response) in a warming climate, because it increases very strongly with increasing temperature and decreasing air humidity, but thereby reflects a response to the surface fluxes rather than a change as a driver of actual evapotranspiration (see also two points above). For instance, Roderick et al. (2015) show that net radiation increases much less than potential evaporation in a warming climate, and ultimately it is the available energy that constrains the surface water fluxes, not Epot itself (Epot is merely a construct to represent this effect). Milly et al. (2016) also provide further discussions on this topic.

→ Again, we fully agree with the reviewer that actual evapotranspiration in dry regions is limited primarily by soil moisture and not by potential evapotranspiration. We wish to emphasize that in our model, the aridity index did not directly drive the results about soil moisture—in agreement with the findings by Gudmundsson et al. (2016) (see our detailed response above). Rather, our simulation experiment is using a model which simulates daily patterns of water cycling, including transpiration and the combined influence of weather, vegetation and soil conditions.

→ We thank the reviewer for the list of interesting articles and the opportunity to add a short description using these citations to the introduction section of our manuscript. This statement clarifies that Epot based approaches are unsuitable for drylands.

Furthermore, effects in marginal systems, such as water-limited systems like drylands, play out as trends or as deviations from normal conditions, i.e., relative rather than absolute changes will be relevant. We also extended the description of the simulation model in the method section by two sentences to explain the approach for simulating transpiration and evaporation rates.

→ We also clarified that we used the FAO/UNEP formulation of the aridity index ($AI = P / E_{pot}$ with $AI \leq 0.5$ indicating drylands) and not the formulation used in the Budyko framework ($AI_b = E_{pot} / P$ with larger values indicating arid zones).

→ We plotted our simulation data according to the outline of Figure 1 by Gudmundsson et al. (2016) where panel (a) shows the sensitivity of $F = ET(\text{actual}) / P$ as a function of aridity $\psi = ET(\text{pot}) / P$. We estimated ω from Fuh's equation and compared the resulting Budyko curve with a locally weighted polynomial regression. The color scheme of our new figure corresponds to Figure 4 of our manuscript (i.e., colors represent regions) and not different values of ω as in the Figure of Gudmundsson et al. (2016). This figure demonstrates that the results of our simulation model reproduce as an emergent feature the relationship of the Budyko framework as presented by (Greve et al. 2015; Gudmundsson et al. 2016).

Based on these issues, and in particular because the authors appear unaware of them, I can unfortunately not recommend publication of this article.

→ Many of these concerns appear to be based on mistaken assumptions about our approach. In fact, our results and our simulation model agree very well with the reviewer (see our detailed responses above). We clarified the text of the manuscript to counteract misunderstandings about our methods and model.

References

- Bagley J, Rosenthal DM, Ruiz-Vera UM, Siebers MH, Kumar P, Ort DR, Bernacchi CJ (2015) The influence of photosynthetic acclimation to rising CO₂ and warmer temperatures on leaf and canopy photosynthesis models. *Global Biogeochemical Cycles*, **29**, 194-206.
- Bradford JB, Schlaepfer DR, Lauenroth WK (2014) Ecohydrology of adjacent sagebrush and lodgepole pine ecosystems: The consequences of climate change and disturbance. *Ecosystems*, **17**, 590-605.

- De Kauwe MG, Medlyn BE, Zaehle S *et al.* (2013) Forest water use and water use efficiency at elevated CO₂ : a model-data intercomparison at two contrasting temperate forest FACE sites. *Glob Chang Biol*, **19**, 1759-1779.
- Feng S, Fu Q (2013) Expansion of global drylands under a warming climate. *Atmospheric Chemistry and Physics*, **13**, 10081-10094.
- Frank DC, Poulter B, Saurer M *et al.* (2015) Water-use efficiency and transpiration across European forests during the Anthropocene. *Nature Climate Change*, **5**, 579-583.
- Greve P, Gudmundsson L, Orlowsky B, Seneviratne SI (2015) Introducing a probabilistic Budyko framework. *Geophysical Research Letters*, **42**, 2261-2269.
- Gudmundsson L, Greve P, Seneviratne SI (2016) The sensitivity of water availability to changes in the aridity index and other factors—A probabilistic analysis in the Budyko space. *Geophysical Research Letters*, **43**, 6985-6994.
- Huang J, Yu H, Guan X, Wang G, Guo R (2015) Accelerated dryland expansion under climate change. *Nature Climate Change*, **6**, 166-171.
- Huang M, Piao S, Zeng Z *et al.* (2016) Seasonal responses of terrestrial ecosystem water-use efficiency to climate change. *Glob Chang Biol*, **22**, 2165-2177.
- Lauenroth WK, Bradford JB (2006) Ecohydrology and the partitioning AET between transpiration and evaporation in a semiarid steppe. *Ecosystems*, **9**, 756-767.
- Medlyn BE, Zaehle S, De Kauwe MG *et al.* (2015) Using ecosystem experiments to improve vegetation models. *Nature Climate Change*, **5**, 528-534.
- Morgan JA, LeCain DR, Pendall E *et al.* (2011) C4 grasses prosper as carbon dioxide eliminates desiccation in warmed semi-arid grassland. *Nature*, **476**, 202-205.
- Morgan JA, Mosier AR, Milchunas DG, LeCain DR, Nelson JA, Parton WJ (2004) CO₂ enhances productivity, alters species composition, and reduces digestibility of shortgrass steppe vegetation. *Ecological Applications*, **14**, 208-219.
- Mueller KE, Blumenthal DM, Pendall E *et al.* (2016) Impacts of warming and elevated CO₂ on a semi - arid grassland are non - additive, shift with precipitation, and reverse over time. *Ecology Letters*, **19**, 956-966.
- Norby RJ, De Kauwe MG, Domingues TF *et al.* (2016) Model–data synthesis for the next generation of forest free-air CO₂ enrichment (FACE) experiments. *New Phytologist*, **209**, 17-28.
- Norby RJ, Zak DR (2011) Ecological Lessons from Free-Air CO₂ Enrichment (FACE) Experiments. *Annual Review of Ecology, Evolution, and Systematics*, **42**, 181-203.
- Palmquist K, Schlaepfer DR, Lauenroth WK, Bradford JB (2016) Mid-latitude shrub steppe plant communities: Climate change consequences for soil water resources. *Ecology*, na/na.
- Pugh TAM, Müller C, Arneth A, Haverd V, Smith B (2016) Key knowledge and data gaps in modelling the influence of CO₂ concentration on the terrestrial carbon sink. *Journal of Plant Physiology*.
- Reich PB, Hobbie SE, Lee TD (2014) Plant growth enhancement by elevated CO₂ eliminated by joint water and nitrogen limitation. *Nature Geoscience*, **7**, 920-924.
- Schlaepfer DR, Ewers BE, Shuman BN, Williams DG, Frank JM, Massman WJ, Lauenroth WK (2014) Terrestrial water fluxes dominated by transpiration: Comment. *Ecosphere*, **5**, art61.

- Schlaepfer DR, Lauenroth WK, Bradford JB (2012) Ecohydrological niche of sagebrush ecosystems. *Ecohydrology*, **5**, 453-466.
- Schymanski SJ, Roderick ML, Sivapalan M (2015) Using an optimality model to understand medium and long-term responses of vegetation water use to elevated atmospheric CO₂ concentrations. *AoB Plants*, **7**.
- Smith NG, Dukes JS (2013) Plant respiration and photosynthesis in global-scale models: incorporating acclimation to temperature and CO₂. *Global Change Biology*, **19**, 45-63.
- Walker AP, Hanson PJ, De Kauwe MG *et al.* (2014) Comprehensive ecosystem model-data synthesis using multiple data sets at two temperate forest free-air CO₂ enrichment experiments: Model performance at ambient CO₂ concentration. *Journal of Geophysical Research: Biogeosciences*, **119**, 937-964.
- Zaehle S, Medlyn BE, De Kauwe MG *et al.* (2014) Evaluation of 11 terrestrial carbon-nitrogen cycle models against observations from two temperate Free-Air CO₂ Enrichment studies. *New Phytologist*, **202**, 803-822.

Reviewers' comments:

Reviewer #3 (Remarks to the Author):

Overall recommendation: minor to major revisions

General comments:

The authors have substantially improved and clarified the manuscript. The article would be suitable for a specialist journal, however, I evaluate that for the Nature Communications audience, there are still a few remaining issues that would need to be addressed before publication. I hesitate to term these last revisions major, however they are important.

Main points:

a) The authors are computing their projections for the RCP8.5 and RCP4.5 scenarios. In the context of the 2015 Paris agreement, however, it is well possible that these scenarios will not be realised. Nonetheless, the authors often use the term "future change", or the future tense "... will..." rather than clarifying that these are projected changes for given emissions scenarios. The authors should clarify this point in the manuscript and they should improve the language so that it is clear that the simulated changes are projections, not predicted changes. I have indicated a few examples in my detailed comments below.

Also, this might lie outside the scope of what the authors can perform at present, but it would be useful for the climate research community if the authors could also assess the results they would obtain for the RCP2.6 scenario, i.e. the only emissions scenario keeping global temperature increase with high probability below 1.5° global warming. If the authors were able to assess whether this scenario substantially decreases the odds of the changes projected with the RCP4.5 and RCP8.5 scenarios, this would be a very important information for the climate research community and the public, and would substantially increase the relevance of this article for Nature Communications. I assess that this inclusion would substantially increase the impact of the article.

b) There are some remaining issues with the cited literature. One main result of the authors' analysis is related to projected changes in soil moisture drought. However, much of the literature they cite is related to changes in aridity, not in soil moisture, although there are several existing articles assessing projected changes in soil moisture, e.g.:

- Burke and Brown, 2008, J. Hydrometeorology
- Orłowsky and Seneviratne 2013, HESS

These two articles emphasise the uncertainty of projected changes in soil moisture (and meteorological) drought, and should be referenced in the introduction (see detailed comments #1 and #2).

Detailed comments:

1) page 2, lines 53-54: "Global climate models (GCMs) predict increases of both climatological aridity and meteorological droughts during the 21st century".

This is true for climatological aridity but not for meteorological droughts. It should be clarified here that the projections of changes in the occurrence of meteorological droughts are uncertain in most regions, and only display robust increases in a few regions (e.g. Burke and Brown 2008, Orłowsky and Seneviratne 2013, HESS). This is also consistent with the results of the authors, which show a high model dependency of the projected change patterns (see supplementary information).

2) page 3, lines 65-67: "The predicted global drying trend is largely robust to variation among models, data sources, and drought indices,". Again, this sentence is incorrect. The aridity trends are consistent, but not the drying trends. Also the trends are "projected" not "predicted".

Please replace the sentence as follows: "The projected global trends towards increased aridity is largely robust to variation among models and data sources, even though potential evapotranspiration by itself is unsuitable for understanding drying trends^{9,18,19}. On the other hand, trends in meteorological drought and soil moisture have been shown to be highly uncertain in most regions and generally model dependent (Burke and Brown 2008, Orlowsky and Seneviratne 2013)."

3) page 4, lines 84-87: Replace "predictions" with "projections". Note that article #23 is for present climate, not for projections, and that article #20 is not analysing soil moisture projections. Burke and Brown (2008) and Orlowsky and Seneviratne (2013) could be cited here.

4) page 6, line 137: representative concentration pathway (RCP) 8.5: Mention that this is a "business as usual" scenario (i.e. no mitigation), which will not occur if the Paris agreement's decision to keep global temperature increase "well below 2°" is realised.

5) page 6, line 141: "will be converted": replace with "would be converted under the considered scenario"

6) page 6, line 143: "will be added": replace with "would be added"

7) pages 16-17: Discussion of emission scenarios (RCP8.6, RCP4.5): Also mention Paris agreement and its possible implications. I would also strongly recommend that the authors include additional analyses for the RCP2.6 scenario.

8) page 18, line 419: Replace "predicted" with "projected"

References:

Burke, E.J., and S. Brown, 2008: Evaluating uncertainties in the projection of future drought. *J. Hydrometeorology*, 9, 292-299.

Orlowsky, B., and S.I. Seneviratne, 2013: Elusive drought: uncertainty in observed trends and short- and long-term CMIP5 projections. *Hydrol. Earth Syst. Sci.*, 17, 1765–1781, 2013, doi:10.5194/hess-17-1765-2013

Detailed responses to reviewer's comments

Reviewer #3 (Remarks to the Author):

Review of Schlepper et al., submitted to Nature Communications

Overall recommendation: minor to major revisions

General comments:

The authors have substantially improved and clarified the manuscript. The article would be suitable for a specialist journal, however, I evaluate that for the Nature Communications audience, there are still a few remaining issues that would need to be addressed before publication. I hesitate to term these last revisions major, however they are important.

Main points:

a) The authors are computing their projections for the RCP8.5 and RCP4.5 scenarios. In the context of the 2015 Paris agreement, however, it is well possible that these scenarios will not be realised. Nonetheless, the authors often use the term "future change", or the future tense "... will..." rather than clarifying that these are projected changes for given emissions scenarios. The authors should clarify this point in the manuscript and they should improve the language so that it is clear that the simulated changes are projections, not predicted changes. I have indicated a few examples in my detailed comments below.

Also, this might lie outside the scope of what the authors can perform at present, but it would be useful for the climate research community if the authors could also assess the results they would obtain for the RCP2.6 scenario, i.e. the only emissions scenario keeping global temperature increase with high probability below 1.5° global warming. If the authors were able to assess whether this scenario substantially decreases the odds of the changes projected with the RCP4.5 and RCP8.5 scenarios, this would be a very important information for the climate research community and the public, and would substantially increase the relevance of this article for Nature Communications. I assess that this inclusion would substantially increase the impact of the article.

→ We clarified in the manuscript that our results are projections based on simulation models and that we do not assume the projections reflect exactly what will happen. We clarified this with more careful use of language throughout the manuscript.

→ Unfortunately, it is not possible at this late stage to perform another major set of new simulation runs under RCP2.6. We also argue that we already present results under two scenarios: an intermediate (RCP4.5) and a high emissions scenario (RCP8.5) to explore the uncertainty with the changes we have to expect, encapsulated by different scenarios -- as has been done frequently in the literature on impacts of climate change (maybe versus the literature on climate and climate change itself): examples of recent high-profile publications which examined impacts under two scenarios RCP4.5 and RCP8.5 or only under one scenario RCP8.5: (Burrows *et al.* 2014; Yoon *et al.* 2015; Duan *et al.* 2016; Milly *et al.* 2016). Clearly, one could argue that a low emissions scenario (RCP2.6) would also be interesting and further scenarios, e.g., RCP6.0 or a

scenario, which reflects the Paris agreements directly -- GCM simulations of the latter currently are not available. Our opinion is that this is unlikely to happen soon, and that the expected changes could roughly be guessed by interpolating current climate and the projections for RCP4.5 and RCP8.5. Furthermore, RCP2.6 forms the low end of the scenario literature in terms of emissions and radiative forcing and suggests even negative emissions from energy use in the second half of the 21st century. Cumulative emissions of greenhouse gases from 2010 to 2100 would need to be reduced by 70% compared to a baseline scenario, which is not realistic even if the 2015 Paris Agreement is fully implemented and all its targets fulfilled. We updated the manuscript to better highlight the uncertainty in emission pathways by RCP4.5 and RCP8.5 and we make it clear that we are talking about climate change projections for these scenarios.

Burrows MT, Schoeman DS, Richardson AJ et al. (2014) Geographical limits to species-range shifts are suggested by climate velocity. *Nature*, 507, 492-495.

Duan K, Sun G, Sun S et al. (2016) Divergence of ecosystem services in U.S. National Forests and Grasslands under a changing climate. *Scientific Reports*, 6, 24441.

Milly PCD, Dunne KA (2016) Potential evapotranspiration and continental drying. *Nature Climate Change*, advance online publication.

Yoon JH, Wang SY, Gillies RR, Kravitz B, Hippias L, Rasch PJ (2015) Increasing water cycle extremes in California and in relation to ENSO cycle under global warming. *Nature Communications*, 6, 8657.

b) There are some remaining issues with the cited literature. One main result of the authors' analysis is related to projected changes in soil moisture drought. However, much of the literature they cite is related to changes in aridity, not in soil moisture, although there are several existing articles assessing projected changes in soil moisture, e.g.:

- Burke and Brown, 2008, *J. Hydrometeorology*

- Orlowsky and Seneviratne 2013, *HESS*

These two articles emphasise the uncertainty of projected changes in soil moisture (and meteorological) drought, and should be referenced in the introduction (see detailed comments #1 and #2).

→ We fixed the cited literature and added the two suggested citations where appropriate.

Detailed comments:

1) page 2, lines 53-54: "Global climate models (GCMs) predict increases of both climatological aridity and meteorological droughts during the 21st century".

This is true for climatological aridity but not for meteorological droughts. It should be clarified here that the projections of changes in the occurrence of meteorological droughts are uncertain in most regions, and only display robust increases in a few regions (e.g. Burke and Brown 2008, Orlowsky and Seneviratne 2013, *HESS*). This is also consistent with the results of the authors, which show a high model dependency of the projected change patterns (see supplementary

information).

→ We clarified the text that GCMs project consistent climatological aridity and display a large uncertainty how meteorological drought will change. We added the two suggested citations.

2) page 3, lines 65-67: "The predicted global drying trend is largely robust to variation among models, data sources, and drought indices,". Again, this sentence is incorrect. The aridity trends are consistent, but not the drying trends. Also the trends are "projected" not "predicted".

→ We replaced 'predict*' with 'project*' here and throughout the manuscript where appropriate.

Please replace the sentence as follows: "The projected global trends towards increased aridity is largely robust to variation among models and data sources, even though potential evapotranspiration by itself is unsuitable for understanding drying trends^{9,18,19}. On the other hand, trends in meteorological drought and soil moisture have been shown to be highly uncertain in most regions and generally model dependent (Burke and Brown 2008, Orlowsky and Seneviratne 2013)."

→ We replaced the sentence as suggested.

3) page 4, lines 84-87: Replace "predictions" with "projections". Note that article #23 is for present climate, not for projections, and that article #20 is not analysing soil moisture projections. Burke and Brown (2008) and Orlowsky and Seneviratne (2013) could be cited here.

→ We replaced 'predict*' with 'project*' here and throughout the manuscript where appropriate.

→ We removed articles #20 and #23 and added the two suggested citations.

4) page 6, line 137: representative concentration pathway (RCP) 8.5: Mention that this is a "business as usual" scenario (i.e. no mitigation), which will not occur if the Paris agreement's decision to keep global temperature increase "well below 2°" is realised.

→ We added this explanation as suggested.

5) page 6, line 141: "will be converted": replace with "would be converted under the considered scenario"

→ We replaced this phrase as suggested.

6) page 6, line 143: "will be added": replace with "would be added"

→ We replaced this phrase as suggested.

7) pages 16-17: Discussion of emission scenarios (RCP8.6, RCP4.5): Also mention Paris agreement and its possible implications. I would also strongly recommend that the authors include additional analyses for the RCP2.6 scenario.

→ We explained RCPs in relation to the Paris agreement and the potential implications for how realistic our results may be under different scenarios.

→ Please see our full response to the request to add RCP2.6 simulations in our response to the first comment.

8) page 18, line 419: Replace "predicted" with "projected"
→ We replaced this word as suggested.

References:

Burke, E.J., and S. Brown, 2008: Evaluating uncertainties in the projection of future drought. *J. Hydrometeorology*, 9, 292-299.

Orlowsky, B., and S.I. Seneviratne, 2013: Elusive drought: uncertainty in observed trends and short- and long-term CMIP5 projections. *Hydrol. Earth Syst. Sci.*, 17, 1765–1781, 2013, doi:10.5194/hess-17-1765-2013

REVIEWERS' COMMENTS:

Reviewer #3 (Remarks to the Author):

The authors have satisfactorily addressed my comments.

Point-by-point response to referees' comments

REVIEWERS' COMMENTS

Reviewer #3:

The authors have satisfactorily addressed my comments.

→ We appreciate the inputs from the reviewers' comments, which have substantially strengthened our manuscript.